# A modular chemoenzymatic cascade strategy for the structure-customized assembly of ganglioside analogs

Xuefeng Jin[1,2,6], Hanchao Cheng[3,4,6], Xiaohui Chen[1], Xuefeng Cao[5], Cong Xiao[5], Fengling Ding[5], Huirong Qu[5], Peng George Wang[4], Yan Feng[1] & Guang-Yu Yang [1✉]

Gangliosides play vital biological regulatory roles and are associated with neurological system diseases, malignancies, and immune deficiencies. They have received extensive attention in developing targeted drugs and diagnostic markers. However, it is difficult to obtain enough structurally defined gangliosides and analogs especially at an industrial-relevant scale, which prevent exploring structure-activity relationships and identifying drug ingredients. Here, we report a highly modular chemoenzymatic cascade assembly (MOCECA) strategy for customized and large-scale synthesis of ganglioside analogs with various glycan and ceramide epitopes. We typically accessed five gangliosides with therapeutic promising and systematically prepared ten GM1 analogs with diverse ceramides. Through further process amplification, we achieved industrial production of ganglioside GM1 in the form of modular assembly at hectogram scale. Using MOCECA-synthesized GM1 analogs, we found unique ceramide modifications on GM1 could enhance the ability to promote neurite outgrowth. By comparing the structures with synthetic analogs, we further resolved the problem of contradicting descriptions for GM1 components in different pharmaceutical documents by reinterpreting the exact two-component structures of commercialized GM1 drugs. Because of its applicability and stability, the MOCECA strategy can be extended to prepare other glycosphingolipid structures, which may pave the way for developing new glycolipid drugs.

[1] State Key Laboratory of Microbial Metabolism, Joint International Research Laboratory of Metabolic & Developmental Sciences, School of Life Sciences and Biotechnology, Shanghai Jiao Tong University, Shanghai, China. [2] Department of Clinical Pharmaceutics, The People's Hospital of Guangxi Zhuang Autonomous Region, Nanning, China. [3] School of Food and Drug, Shenzhen Polytechnic University, Shenzhen, China. [4] Department of Pharmacology, Key University Laboratory of Metabolism and Health of Guangdong, School of Medicine, Southern University of Science and Technology, Guangdong, China. [5] Glycogene LLC, 10th Floor, Building 3, Wuhan Precision Medicine Industrial Base, East Lake New Technology Development Zone, Wuhan, China. [6] These authors contributed equally: Xuefeng Jin, Hanchao Cheng. ✉email: yanggy@sjtu.edu.cn

Gangliosides are composed of sialic acid-containing oligo-saccharide chains attached to ceramides, which are composed of sphingosine and fatty acid linked by an amide bond. Gangliosides are distributed in the membranes of vertebrate cells[1,2], which perform important regulatory roles in multiple major human diseases including nervous system diseases[3,4], lysosomal storage diseases[5], immune system diseases[6], and cancers[7]. They have been explored as biomarkers for clinical diagnosis and as antigenic targets for disease treatment[8–13].

Gangliosides possess diverse glycan epitopes and show various pathophysiological characteristics. The biological activities of GM1 in neurotrophy and neuroprotection have been applied to therapeutic research for central nervous system injury and Parkinson's disease[14]. The aggregation and precipitation of GM2 in lysosome of special brain regions is the typical pathological feature of Tay Sachs, a sphingolipid lysosomal storage disorder[15,16]. GM3, GD2, and GD3 have been investigated as tumor-associated antigens[9,17,18], which are widely used for developing therapeutic antibodies, drug conjugates, and chimeric antigen receptor T cells. Recent studies characterizing the biological effects of gangliosides have revealed that ganglioside functions are also regulated by the ceramide structure. For example, GM1 is known to have neuroprotective effects in the mammalian brain; however, a GM1 analogs with a C18:3 substitution or deletion of fatty acids induced apoptosis of Neuro2a cells[19]. The roles of GM3 and GD3 as endogenous activators of invariant natural killer T cells were found to be heavily reliant on their ceramide structures[20]. The ratios of C20/C18 fatty acids and d20/d18 sphingosines in the human cerebral cortex and white matter were shown to gradually increase with aging, suggesting that the change of ceramides in gangliosides is related to the development and degeneration of nervous system[21]. Elucidating the functions of ganglioside analogs will require the isolation of single structures; however, because of the microheterogeneity of natural gangliosides and, especially, the complexity and rarity of gangliosides composed of unique ceramides in cells, it is very difficult to obtain sufficient quantities of single analogs by extraction. This hampers the widespread application of gangliosides in medicinal chemistry.

The development of strategies to obtain single components of various gangliosides has gained great interest from synthetic chemists and biologists. Multiple gangliosides, including GM1, GD3, and GD2, have been synthesized chemically with complex operations[22]. But formidable challenges remain in the chemical synthesis of gangliosides because of the regio- and stereo-selectivities of sialylation and insufficient coupling of oligosaccharides with lipids. Using chemoenzymatic technology, Chen et al. enable more effective synthesis for several extended gangliosides[23]. Despite the progress in the synthesis of gangliosides using chemical and chemoenzymatic methods, most research has been focused on the preparation of gangliosides with different glycosyl groups. Little attention has been paid to exploring the diversity of ceramides. Gangliosides with different length sphingosine modifications have not been effectively constructed as D-ribo-phytosphingosine is generally used as a precursor[24,25]. Moreover, previous synthesis attempts only produced gangliosides at a milligram scale[1,23]. It is very necessary to develop an effective method to synthesize gangliosides on a more industrially relevant scale. An effective technology to obtain structurally well-defined ganglioside analogs with various sphingosines and fatty acids modification is worth developed for thoroughly understanding their pharmacological activities.

As a kind of gangliosides used in clinic, commercialized GM1 drugs named Sygen™ from natural extraction was prescribed for vascular or traumatic central nervous system injury and Parkinson's disease and continuously used in China[4]. Due to the microheterogeneity and rarity of gangliosides with unique ceramides by extraction, the structure-activity relationship of various GM1 components in commercial drugs has not been clarified, which may result in potential risks for ganglioside drugs in terms of safety and indication. Commercially available GM1 drug with neuroprotective effects is composed of two analogs. What surprised us particularly is that there are widespread and universal contradictions in descriptions of GM1 components in different fields. Commercialized GM1 drug is annotated to be a two-component mixture of GM1 with C18:0 and C20:0 saturated fatty acids in various pharmaceutical documents. These structural annotations contradict the reports by academic researchers that GM1 in mammalian brain such as human[21], bovine[26] and murine[27,28] mainly consists of d18:1 and d20:1 sphingosines. As commercialized GM1 drugs are mainly extracted from pig brains, it remains unresolved whether the GM1 composition varies among different species or there is a misunderstanding about the drug structure. It is necessary and urgently needed to obtain single structures of GM1 analogs for revealing biological functions and identifying drug components.

Here, we describe a highly modular chemoenzymatic cascade assembly (MOCECA) strategy to achieve the customized, large-scale synthesis of single gangliosides and related analogs by precisely regulating the combination of various oligosaccharides, sphingosines, and fatty acids. The assembly of analogs is now more diversified and convenient for industrialization thanks to the hectogram and high-purity preparation of various modules. Using the MOCECA strategy, we represently generated several gangliosides with promise for therapeutic use and GM1 analogs with diverse ceramides. Based on MOCECA-synthesized GM1 analogs, unique ceramide modifications on GM1 were proved to have different neurobiological activities, and importantly, we reinterpreted the exact two-component structure of GM1 commercialized drug. Our findings demonstrate that the MOCECA strategy is a valuable tool for developing novel ganglioside drugs, which can be applied for exploring structure-activity relationships and identifying drug ingredients.

## Results and discussion

**The MOCECA strategy**. The MOCECA strategy was designed to perform customized and large-scale synthesis of gangliosides and analogs via structural assembly of an oligosaccharide, sphingosine, and fatty acid (Fig. 1). This strategy consisted of four modules: (1) Generation of D-sphingosines with diverse lengths by chemical synthesis with high purity and a high diastereomeric excess (de) value of more than 99%. (2) Preparation of various oligosaccharide fluorides through multiple one-pot enzymatic glycosylation reactions of lactose fluoride (Lac-F). (3) Synthesis of glycosylsphingosines via enzymatic assembly of oligosaccharide fluorides and sphingosines with a yield of more than 84% using glycosynthases[29,30]. (4) Assembly of fatty acids on the glycosyl-sphingosines using the *Shewanella alga G8* sphingolipid ceramide N-deacylase (SA_SCD) enzymatic synthesis system. These four modules all have excellent characteristics including cheaply available initial materials, high yields, and good compatibility with a variety of modified substrates, which makes MOCECA highly flexible. We present examples for each module to illustrate the application of MOCECA strategy.

**Scalable and cost-effective preparation of D-sphingosines**. Sphingosine and its analogs are important structural and functional components of cell processes, and they play regulatory roles in cell growth, differentiation, and apoptosis[31]. One of the current methods uses D-ribo-phytosphingosine to solve the chirality of 3-OH and prepare D-sphingosine (d18:1) (Fig. 2a), but such strategies are unable to generate sphingosine analogs

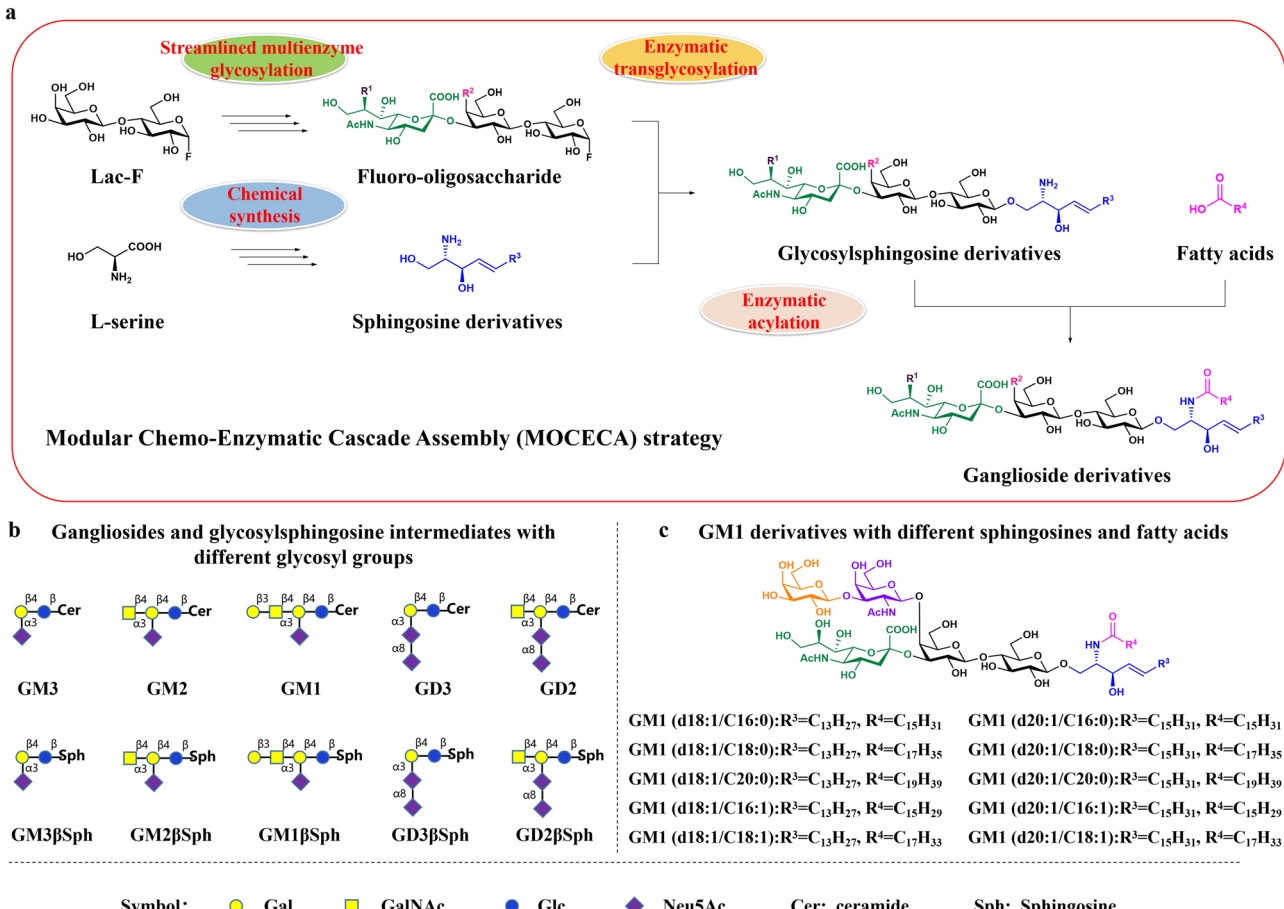

**Fig. 1 The MOCECA strategy for customized synthesis of ganglioside analogs and glycosylsphingosine intermediates. a**, The complete structurally controllable assembly routes and various catalytic modules of MOCECA. Sphingosine analogs produced by chemical synthesis and glycosylsphingosine analogs synthesized by enzymatic transglycosylation needs to be purified for the next reaction. **b**, MOCECA-synthesized gangliosides and glycosylsphingosine intermediates with different glycosyl groups. **c**, MOCECA-synthesized GM1 analogs with different ceramides.

with different lengths[32]. Utilizing other technologies to fabricate sphingosine, such as serine derivatization or synthesis with carbohydrate precursors, requires intricate protection and deprotection steps, resulting in low yield and poor economy[33].

We adopt an improved chemical method to generate high-purity sphingosine analogs with diverse chain lengths. D-sphingosine (d18:1) and D-sphingosine (d20:1) were prepared efficiently from inexpensive L-serine on a large scale to supply enough starting materials for assembly of ganglioside analogs (Fig. 2). L-serine was treated with methanol and di-*tert*-butyl dicarbonate to protect the carboxyl and amine groups, producing compound **28** with 98% yield. Then, in the presence of imidazole, **28** was reacted with *tert*-butyldimethylsilyl chloride (TBDMSCl) to obtain compound **29** for equivalent conversion. Compound **29** was processed with dimethoxy methyl phosphine oxide and n-BuLi at −78 °C to create **30** with 93% yield. We next coupled **30** with *n*-tetradecanal and *n*-hexadecaldehyde to generate compounds **31** and **32**, both with 81% yield. For the following step, selective reduction of ketone, we used LiAlH(O$^t$Bu)$_3$, which shows good selectivity, as the reducing reagent to obtain compounds **33** and **34** (ee>99%), both with 95% yield. Compounds **33** and **34** were treated with 1 M HCl in methanol to generate compounds **35** and **36** (both with 99% yield), which were then treated with acetyl chloride in methanol to obtain D-sphingosines with different chain lengths (60% yield). Using this optimized synthetic method, high-purity D-sphingosine (d18:1) (**16**) and D-sphingosine (d20:1) (**17**) could easily be obtained in seven steps at a low cost and at a hectogram scale, with an overall yield of 41% (Fig. 2b). The structural identification of D-sphingosine (d18:1) shows that it is consistent with the reported compound structure[25]. Compared with a previous method in which sphingosine was generated through the selective reduction of ketone to form a chiral 3-OH[23,34–36], this enhanced synthetic route exhibited superior stereoselectivity and purity. Compared with another approach starting from D-ribo-phytosphingosine, which is restricted to generation of D-sphingosine (d18:1)[24,25], our method to obtain various sphingosine modules is more versatile. In addition, our synthetic process makes it easy to produce designated-length sphingosines in accordance with actual needs through technical optimization.

## Synthesis of fluoro-oligosaccharides through streamlined multi-enzymatic glycosylation.

To facilitate the synthesis of targeted gangliosides, several types of fluoro-oligosaccharides were synthesized using four different one-pot multi-enzymatic (OPME) modules (Fig. 3). Lac-F (1.38 g) was used as the initial acceptor substrate for O-linked α-2,3 sialylation catalyzed by *Neisseria meningitidis* CMP-sialic acid synthetase (NmCSS)[37] (52 mg), *Pasteurella multocida* α-2,3-sialyltransferase 1 (PmST1)[38] (50 mg), and *P. multocida* inorganic pyrophosphatase (PmPpA)[39] (44 mg). Trisaccharide GM3-F was readily synthesized with 95% yield at 37 °C for 4.5 h. The following GM and GD series of fluoro-oligosaccharides were synthesized via two different catalytic routes with GM3-F as a substrate.

**Fig. 2 Chemical synthesis of D-sphingosine (d18:1) and D-sphingosine (d20:1) on a large scale. a**, Structures of D-ribo-phytosphingosine and D-sphingosine (d18:1). **b**, Modified synthesis route of D-sphingosine (d18:1) and D-sphingosine (d20:1) from L-serine (200.0 g, 1.903 mol). Reagents and conditions: (i) methanol, AcCl (5.415 mol, 2.845 equiv), 80 °C, 2 h; (ii) (Boc)$_2$O (2.144 mol), TEA (4.266 mol), CH$_2$Cl$_2$, 20–30 °C overnight, 98% yield in two steps; (iii) imidazole (3.143 mol, 1.685 equiv), TBDMSCl (3.118 mol, 1.672 equiv), CH$_2$Cl$_2$, 20–30 °C overnight, **29**, quantitative yield; (iv) DMMP (6.407 mol, 3.425 equiv), THF, n-BuLi (2.5 M in hexane, 6.300 mol), −75 °C, 1 h, **30**, 93% yield; (v) n-tetradecanal (1.460 mol) or n-hexadecaldehyde (1.460 mol), TEA (2.892 mol), LiCl (2.854 mol, 1.646 equiv), THF, 15–20 °C, 5 h, **31**: 81% yield, **32**: 81% yield; (vi) LiAlH(O$^t$Bu)$_3$ (2.997 mol, 2.130 equiv), ethanol, −78 °C, 2 h, **33**: 95% yield, **34**: 95% yield; (vii) HCl, methanol, 25 °C, 17 h, **35**: 99% yield, **36**: 99% yield; (viii) AcCl (7.314 mol), methanol, 25 °C, 15 h, **16**: 60% yield, **17**: 60% yield. Abbreviations: AcCl, Acyl chloride; (Boc)$_2$O, di-tert-butyl dicarbonate; TEA, triethylamine; TBDMSCl, tert-butyldimethylsilylchloride; THF, tetrahydrofuran; DMMP, dimethyl methyl phosphonate (see more details in Supplementary II).

**Route 1 for generation of GM2-F and GM1-F**. The N-acetylgalactosamine (GalNAc) OPME glycosylation system consisting of *Bifidobacteriumlongum* strain ATCC55813 N-acetylhexosamine-1-kinase (BLNahK)[40] (40 mg), human UDP-GalNAc pyrophosphorylase (AGX1)[41] (30 mg), PmPpA (20 mg), and *Campylobacter jejuni* β1-4GalNAcT (CjCgtA)[42] (100 mg) was used to glycosylate GM3-F to generate GM2-F. After adding GalNAc to the β1-4 linkage of the substrate, tetrasaccharide GM2-F was obtained with 99% yield in 3 h. With GM2-F in solution as a substrate, the OPME galactose (Gal) glycosylation system consisting of *C. jejuni* β1-3-galactosyltransferase (CjCgtB)[43] (120 mg), *Escherichia coli* galactokinase (EcGalK)[44] (60 mg), *Arabidopsis thaliana* UDP-sugar pyrophosphorylase (AtUSP)[45] (52 mg), and PmPpA (20 mg) was used to generate GM1-F pentasaccharides. By adding a β1-3-linked galactoside on GalNAc, GM1-F was obtained with a yield of 98% in 6 h. It is worth noting that an excessive amount of Gal in the presence of CjCgtB would react with GM1-F via formation of a β1-3 linkage of Gal to Gal to generate the by-product Gal-GM1-F. Through parameter optimization, the molar ratio of Lac-F to Gal was controlled at 1:1.1 to produce only the desired product.

**Route 2 for generation of GD3-F and GD2-F**. The OPME α2,8 sialylation system consisting of *C. jejuni* α2-3/8 sialyltransferase (CjCstII)[46] (48 mg), NmCSS (60 mg), and PmPpA (52 mg) was used to glycosylate GM3-F to synthetize GD3-F. After adding α2-8 linked N-acetylneuraminic acid (Neu5Ac) to the substrate,

GD3-F was produced with a yield of 70% overnight. Using the OPME GalNAc glycosylation system consisting of BLNahK (40 mg), AGX1 (42 mg), PmPpA (20 mg), and CjCgtA (100 mg), GD2-F was synthesized starting from GD3-F with a yield of 96% in 4 h. Each step of the catalytic reaction was monitored by high-performance liquid chromatography (HPLC) and thin-layer chromatography. The mild reaction conditions and specificity of enzymatic synthesis allowed the whole glycosylation reaction to be completed in two days without elaborate purification steps. Construction, expression and purification of various enzymes see supplementary methods.

The β1,4N-acetylgalactosaminylation system consisted of *Bifidobacteriumlongum* strain ATCC55813 N-acetylhexosamine-1-kinase (BLNahK)[40], human UDP-GalNAc pyrophosphorylase (AGX1)[41], PmPpA, *Campylobacter jejuni* β1-4GalNAcT (CjCgtA)[42], N-acetylgalactosamine (GalNAc), adenosine 5'-triphosphate (ATP), and uridine 5'-triphosphate (UTP).

The β1,3 galactosylation system consisted of *Escherichia coli* galactokinase (EcGalK)[44], *Arabidopsis thaliana* UDP-sugar pyrophosphorylase (AtUSP)[45], PmPpA, *C. jejuni* β1,3galactosyltransferase (CjCgtB)[43], ATP, and UTP.

The α2,8 sialylation system consisted of *C. jejuni* α 2-3/8 sialyltransferase (CjCstII)[46], NmCSS, galactose (Gal), PmPpA, Neu5Ac, and CTP (see more details in Supplementary II).

**Large-scale and structurally controllable synthesis of gangliosides with different glycosyl groups**. With a large quantity of diverse sphingosines and fluoro-oligosaccharides in hand, we explored the synthetic universality of the MOCECA strategy for various gangliosides with different glycosyl groups. The transglycosylation and acylation modules of MOCECA were used to assemble oligosaccharides, sphingosines, and fatty acids. As the ceramide structure of natural gangliosides is mainly composed of D-sphingosine (d18:1) and stearic acid, we carried out modular assembly of the compounds with this component. As *Rhodococcus* strain M-777 endoglycoceramidase II (EGC II) D314Y/E351S was reported to be a glycosphingolipid (GSL) endoglycosidase mutant with transglycosylation activity[30], it was used to catalyze the linkage of fluoro-oligosaccharides and sphingosines in our strategy. A mixed solvent containing 10% ethanol (v/v) instead of Triton-X was used to solubilize sphingosine for convenient purification in the next step. Two equivalents of D-sphingosine (d18:1) and a sufficient amount of enzyme (800 mg) were added to ensure the efficient conversion of oligosaccharide fluorides. After optimizing the catalytic system by adjusting the type of cosolvent, enzyme dosage, and the ratio of substrates, various fluoro-oligosaccharides including GM1-F, GM2-F, GM3-F, GD2-F, and GD3-F were found to directly assemble with D-sphingosine (d18:1) in the presence of EGC-II E351S/D314Y. The whole reaction was incubated at 37 °C for 10 h to obtain the desired glycosylsphingosines. As the EGC-II mutant has relatively relaxed acceptor specificities and catalytic conditions, GM3βSph, GM2βSph, GM1βSph, GD3βSph, and GD2βSph were easily prepared in separate reactions at the multigram scale with product yields ranging from 84%–92% after purification (Table. 1).

Sphingolipid ceramide N-deacylase not only hydrolyzes the amide bond of ceramide in GSLs to produce glycosylsphingosine, but also catalyzes the reverse amidation reaction to generate GSLs. SA_SCD was characterized in our previous study, and we achieved the bidirectional control of its catalytic equilibrium[47,48]. Here, we screened a series of parameters including the ratio of substrates, pH, cosolvents, surfactants, and metal ions to promote the acylation of glycosylsphingosines and obtain gangliosides. Glycosylsphingosines **6–10** were coupled with stearic acid to form the amidation products gangliosides **1–5**

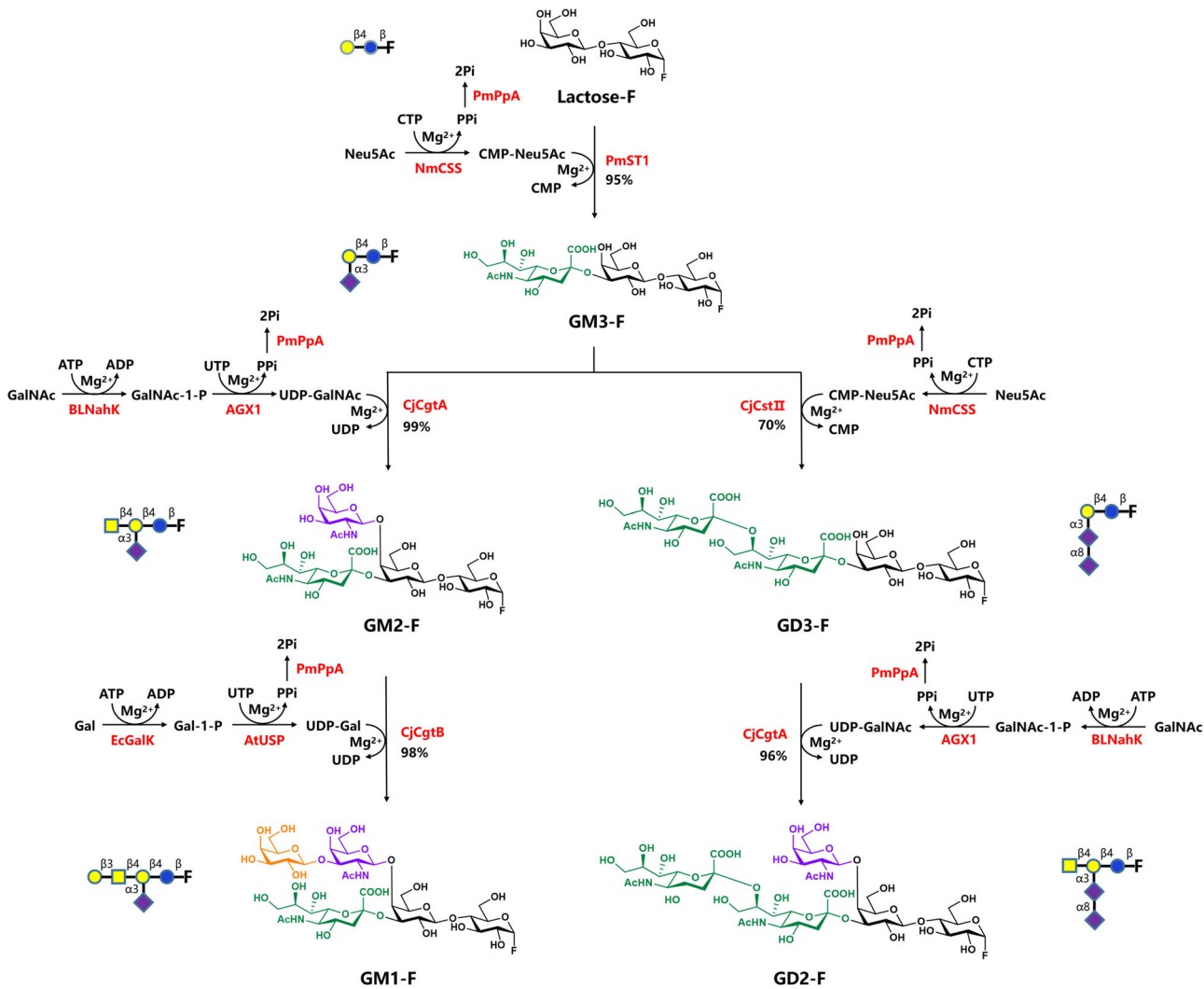

**Fig. 3 Synthesis of fluoro-oligosaccharides using a streamlined multi-enzymatic glycosylation system.** The α2,3 sialylation system consisted of *Neisseria meningitidis* CMP-sialic acid synthetase (NmCSS)[37], *Pasteurella multocida* α2-3-sialyltransferase 1 (PmST1)[38], *P. multocida* inorganic pyrophosphatase (PmPpA)[39], N-acetylneuraminic acid (Neu5Ac), and cytidine 5′-triphosphate (CTP).

in the presence of the SA_SCD (160 mg) synthetic system at 37 °C for 5 h. Five gangliosides with different oligosaccharide chains, namely GM1, GM2, GM, GD2, and GD3, were rapidly prepared with 60%–71% yield (Table 1). The structures of these purified compounds were identified by high resolution mass spectrometry (HRMS) and ¹H and ¹³C NMR (Supplementary Data). Using the MOCECA strategy we have successfully achieved the customized gram-scale synthesis of several important gangliosides in a low-cost and simple manner and confirmed the effectiveness of this strategy for accurate assembly of different glycosyl groups with ceramide. The whole strategy is convenient for process amplification, and it is promising for application in industrial production.

**Structurally controllable and industrial synthesis of GM1 analogs with different sphingosines and fatty acids**. The sphingosines and fatty acids in gangliosides were reported to show different tissue specificities and temporal specificities in the central nervous system, suggesting they may be involved in brain development and ageing[21,27,49]. The MOCECA strategy was next applied to produce GM1 analogs with different sphingosines and fatty acids to test its effectiveness in the assembly of glycosyl groups and diverse ceramide structures. GM1-F obtained from

preliminary purification could directly assemble with d18:1 and d20:1 D-sphingosines in reactions catalyzed by EGC-II E351S/D314Y. GM1βSph (d18:1) and GM1βSph (d20:1) were easily prepared in separate reactions with yields over 90% after purification (Table 2).

GM1βSph (d18:1) or GM1βSph (d20:1) was then reacted with diverse fatty acids including palmitic acid (C16:0), stearic acid (C18:0), arachidic acid (C20:0), palmitoleic acid (C16:1), and oleic acid (C18:1) using the SA_SCD enzymatic system at 37 °C for 5 h. The enzymatic amidation reactions yielded six types of saturated GM1 analogs, namely GM1 (d18:1/C16:0), GM1 (d18:1/C18:0), GM1 (d18:1/C20:0), GM1 (d20:1/C16:0), GM1 (d20:1/C18:0), and GM1 (d20:1/C20:0), with yields ranging from 64%–73%; these six analogs are the main GM1 components in the human brain[21]. In addition, four unsaturated GM1 analogs were synthesized, namely GM1 (d18:1/C16:1), GM1 (d18:1/C18:1), GM1 (d20:1/C16:1), and GM1 (d20:1/C18:1), with yields ranging from 76%–82% (Table 2). The structures of these purified compounds were verified by HRMS and ¹H and ¹³C NMR (Supplementary Data).

Not limited to the integrated synthesis of GM1 analogs at the gram level, we conducted process scale-up experiments using MOCECA strategy. As GM1, which consists of d18:1 sphingosine and C18:0 fatty acid, is naturally present and widely distributed in

**Table 1 Large-scale and structurally controllable synthesis of gangliosides with different glycosyl groups .**

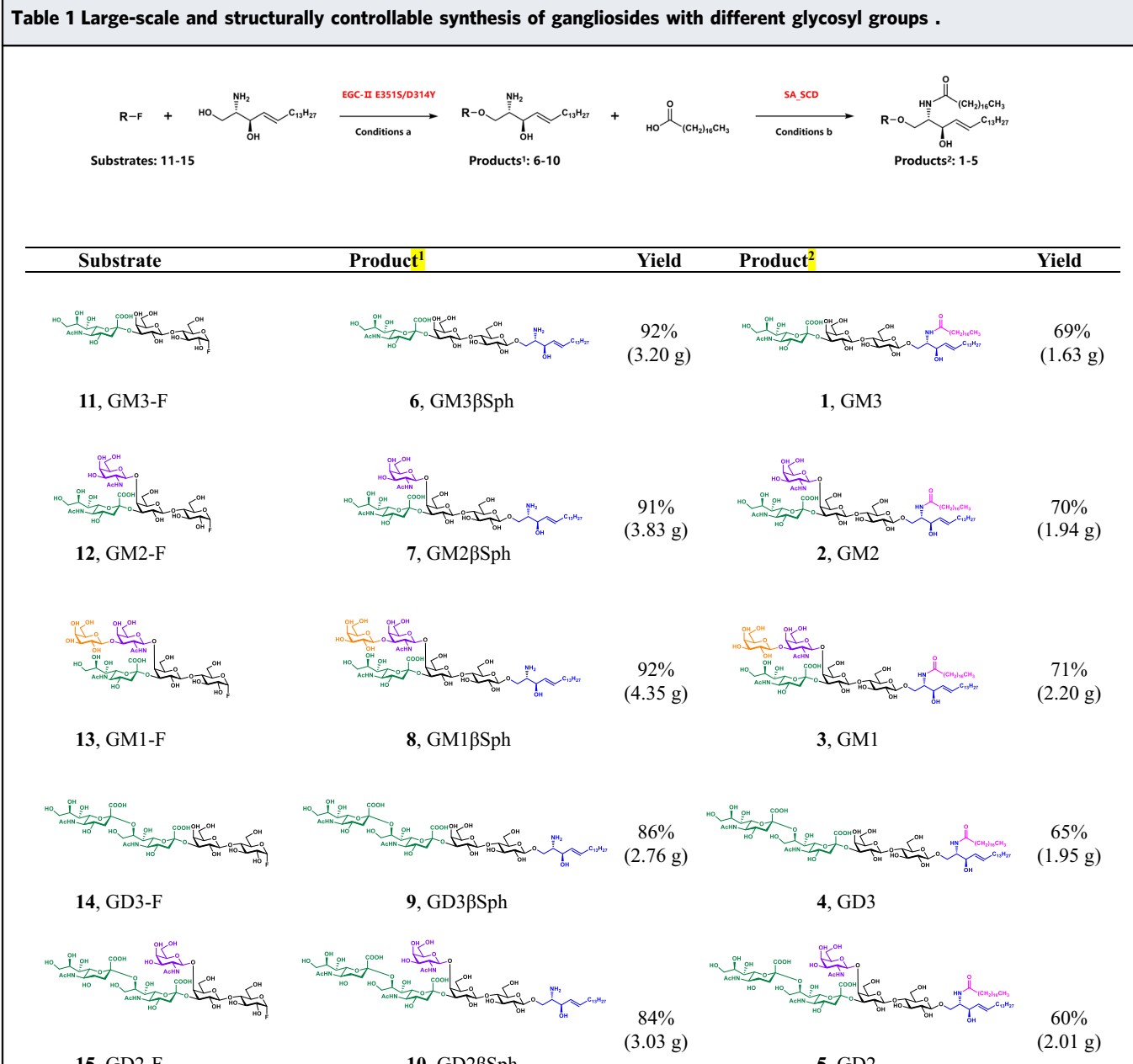

| Substrate | Product[1] | Yield | Product[2] | Yield |
|---|---|---|---|---|
| **11**, GM3-F | **6**, GM3βSph | 92% (3.20 g) | **1**, GM3 | 69% (1.63 g) |
| **12**, GM2-F | **7**, GM2βSph | 91% (3.83 g) | **2**, GM2 | 70% (1.94 g) |
| **13**, GM1-F | **8**, GM1βSph | 92% (4.35 g) | **3**, GM1 | 71% (2.20 g) |
| **14**, GD3-F | **9**, GD3βSph | 86% (2.76 g) | **4**, GD3 | 65% (1.95 g) |
| **15**, GD2-F | **10**, GD2βSph | 84% (3.03 g) | **5**, GD2 | 60% (2.01 g) |

**Condition a**: Substrates **11–15**, substrate **16** (2.40 g, 8.00 mmol), and purified EGC-II E351S/D314Y (800 mg) were incubated in 100 mM NaAc buffer (pH 5.3) with 10% ethanol (v/v) at 37 °C for 10 h. Yield after purification using C18 cartridges is shown.
**Condition b**: Substrates **6–10** (2.00 mmol), stearic acid (853 mg, 3.00 mmol), and purified SA_SCD (160 mg) were incubated in 800 mL of Tris-HCl buffer (50 mM, pH 7.5) with 10% DME (v/v) at 37 °C for 6 h. Yield after purification using Sep Pak VAC C18 cartridges and silica gel filtration column is shown. Abbreviations: EGC-II E351S/D314Y, endoglycoceramidase II E351S/D314Y. SA_SCD, *Shewanella alga* G8 sphingolipid ceramide N-deacylase (see more details in Supplementary II).

mammals, we take GM1 (d18:1/C18:0) as an example for preparation. Chemical synthesis of sphingosines, streamlined multi-enzymatic glycosylation system, transglycosylation reaction of EGC-II mutant and acylation reaction of SA_SCD can achieve the synthesis and assembly of various modules for hectogram-scale industrialization. The yield of GM1 (d18:1/C18:0) synthesized by the MOCECA strategy reached 113.35 g (part II in SI). Hence, customized GM1 analogs with defined structures were successfully obtained by industrialization using the MOCECA strategy, demonstrating the effective catalytic performance of this strategy.

The cost estimation for the preparation of 100 gram of GM1 through MOCECA strategy was conducted. (1) Total cost of main substrates and reagents was 2013.70 USD. (2) Total cost of various enzymes was 37241.93 USD. (3) Labor cost was 5519.68

USD. Purification cost was 2434.79 USD. The cost to synthesize 100 grams of GM1 in total was 47210.10 USD (Average cost was 472.10 USD/g) (Supplementary Table 1). Further engineering studies including immobilization of the enzymes and optimization of the purification steps are feasible for significantly reducing the total cost, which give MOCECA great potential to become a cost-effective technology for large-scale customized synthesis of gangliosides in the future.

**Differences in the neurobiological activities of MOCECA-synthesized GM1 analogs in Neuro2a cells.** Neurite growth, which is an important process in neuron differentiation, development, and maturation, can repair nerve damage caused by

**Table 2 Structurally controllable and integrated synthesis of GM1 analogs with different sphingosines and fatty acids .**

| Substrate[1] | Product[1] | Yield | Substrate[2] | Product[2] | Yield |
|---|---|---|---|---|---|
| | | | | **19**, GM1 (d18:1/C16:0) | 73% (1.38 g) |
| | | | | **3**, GM1 (d18:1/C18:0) | 71% (113.35 g) |
| **16** | **8**, GM1βSph (d18:1) | 92% (130.31 g) | | **20**, GM1 (d18:1/C20:0) | 66% (1.29 g) |
| | | | | **21**, GM1 (d18:1/C16:1) | 82% (1.54 g) |
| | | | | **22**, GM1 (d18:1/C18:1) | 79% (1.51 g) |
| | | | | **23**, GM1 (d20:1/C16:0) | 70% (1.32 g) |
| | | | | **24**, GM1 (d20:1/C18:0) | 69% (1.33 g) |
| **17** | **18**, GM1βSph (d20:1) | 91% (10.98 g) | | **25**, GM1 (d20:1/C20:0) | 64% (1.25 g) |
| | | | | **26**, GM1 (d20:1/C16:1) | 79% (1.49 g) |
| | | | | **27**, GM1 (d20:1/C18:1) | 76% (1.46 g) |

[1]**Condition a**: For industrial preparation of GM1βSph (d18:1), 24 L GM1-F oligosaccharide fluoride, substrates **16** (240 mmol) and purified EGC-II E351S/D314Y (24 g) were incubated in 100 mM NaAc buffer (pH 5.3) with 10% ethanol (v/v) at 37 °C overnight. For GM1βSph (d20:1) preparation at the 10-gram scale, 2 L GM1-F oligosaccharide fluoride, substrates **17** (20 mmol) and purified EGC-II E351S/D314Y (2 g) were incubated in 100 mM NaAc buffer (pH 5.3) with 10% ethanol (v/v) at 37 °C for 10 h. Yield after purification using Sep Pak VAC C18 cartridges is shown.
[2]**Condition b**: For industrial preparation of GM1 (d18:1/C18:0), substrate **8** (101.76 mmol), stearic acid (C18:0) (152.62 mmol) and purified SA_SCD (10 g) were incubated in 50 L of Tris-HCl buffer (10 mM, pH 7.5) with 10% DME (v/v). The whole catalytic system was carried out in a 100 L bioreactor with 120 rpm at 37 °C overnight. 1 M NaOH and 1 M HCl loaded separately into two reservoirs in bioreactor were used for automatic and real-time control of pH value at 7.5. For preparation of other GM1 analogs, substrate **8** or **18** (1.22 or 1.20 mmol), a fatty acid (1.82 or 1.80 mmol of palmitic acid, stearic acid, arachidic acid, palmitoleic acid, or oleic acid), and purified SA_SCD (160 mg) were incubated in 800 mL of Tris-HCl buffer (20 mM, pH 7.5) with 10% DME (v/v) at 37 °C for 6 h. Yield after purification using Sep Pak VAC C18 cartridges and silica gel filtration columns is shown (see details in Supplementary II).

injury or neurodegenerative diseases[50,51]. Neuro2a cells are a classic model used to study the growth and differentiation of nerve cells, and they can be used to detect the neurite growth-promoting activity of drugs[52,53]. After obtaining the afore mentioned 10 single GM1 structures and two intermediate analogs using the MOCECA strategy, we explored their neurobiological activities.

To test the activities of different analogs, control Neuro2a cells were left untreated (control group) or treated with one of the six GM1 analogs with saturated fatty acids, which naturally occur in the human brain, namely GM1 (d18:1/C16:0), GM1 (d18:1/C18:0), GM1 (d18:1/C20:0), GM1 (d20:1/C16:0), GM1 (d20:1/C18:0), and GM1 (d20:1/C20:0). The effects of the treatments on neurite outgrowth activity were assessed by measuring neurite length and the proportion of cells bearing neurites at a GM1 analogs concentration of 25 μM (Fig. 4a). Among the six saturated GM1 analogs, three with d20:1 sphingosine were observed to promote neurite outgrowth (Fig. 4b, c). One of the three saturated GM1 analogs with the d18:1 sphingosine component, GM1 (d18:1/C18:0), had an effect on neurite outgrowth similar to that of d20:1 sphingosine, and the proportion of this ganglioside in the brain was previously shown to decrease with age[21]. GM1 (d18:1/C16:0) and GM1

(d18:1/C20:0) showed relatively weak activities, and there was no statistical difference between these treatment groups and the control group (Fig. 4b, c). A previous study showed that as people grow older, the proportion of GM1 (d18:1/C20:0) increases and that of GM1 (d18:1/C16:0) remains basically unchanged[21]. Therefore, we speculate that the changes in the composition of GM1 analogs with the d18:1 component may be associated with human growth and aging.

As GM1 analogs with a C16:1 or C18:1 fatty acid have specific effects on transmembrane retrograde trafficking[54], we also studied the activities of these unsaturated analogs. The GM1 (d18:1/C16:1), GM1 (d18:1/C18:1), GM1 (d20:1/C16:1), and GM1 (d20:1/C18:1) groups all showed significantly more neurite growth compared with control Neuro2a cells and the GM1 (d18:1/C18:0) group (Fig. 4a–c). We also carried out cell experiments to test the activities of two GM1βSph intermediates. GM1βSph (d18:1) and GM1βSph (d20:1) also increased neurite outgrowth and length (Fig. 4a–c). Taken together, our assays showed that GM1 analogs with diverse ceramide structures have different promoting activities on neurite outgrowth in cells. As we can easily obtain these GM1βSph intermediates at the gram scale, the synthesis strategy can provide sufficient raw materials

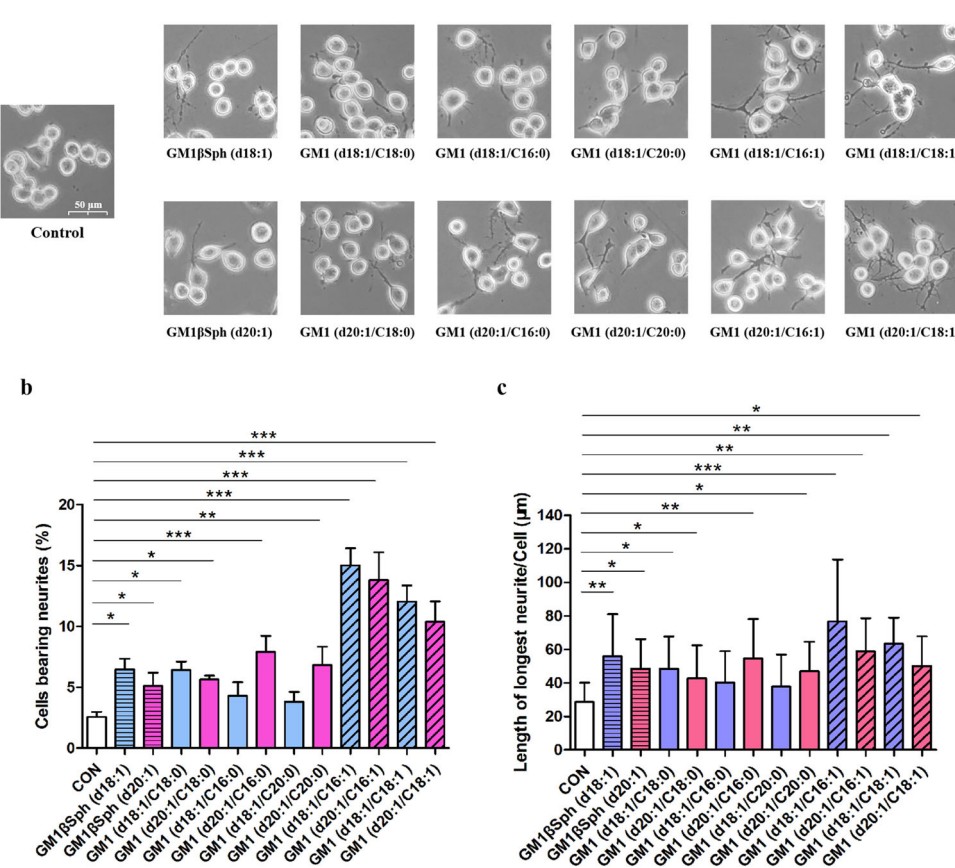

**Fig. 4 Differential neurogenic activities of GM1 and GM1βSph analogs in Neuro2a cells. a** Representative digital images of Neuro2a cells in the control, GM1 analogs, and GM1βSph intermediate groups. **b** Percentages of control Neuro2a cells (CON), cells treated with GM1 analogs (at a concentration of 25 μM), and cells treated with GM1βSph analogs (25 μM) bearing neurites, which were defined as outgrowths more than twice the length of the cell body. **c** The lengths of the longest neurites of CON cells, cells treated with GM1 analogs (25 μM), and cells treated with GM1βSph analogs (25 μM). Neurite lengths were measured morphometrically. Data are presented as the means ± SD of triplicate measurements. GM1 with d18:1 sphingosine were shown in blue or dark blue. GM1 with d20:1 sphingosine were shown in pink or red. GM1βSph analogs were shown by the bar with horizontal lines. Saturated GM1 analogs were labeled by the chart of hollow bar. Unsaturated GM1 analogs are labeled by the bar chart with diagonal line. Significant differences among groups were determined using Newman-Keuls multiple comparison test following one-way ANOVA analysis in GraphPad Prism software. All data are shown as the mean ± SD from three independent experiments. (*$p < 0.05$, **$p < 0.01$, ***$p < 0.001$).

for the preparation of related drugs based on the structures of these intermediates.

## Identification of the components in commercialized GM1 drugs through comparisons with MOCECA-synthesized GM1 analogs.

Commercially available GM1 drug extracted from mammalian brains mainly contain two GM1 analogs. However, there are universal contradictions in descriptions these extracts. Commercialized GM1 drug is annotated to be a two-component mixture of GM1 contained C18:0 and C20:0 saturated fatty acids in the national drug standard formulated by China Food and Drug Administration (CFDA), the patent application for GM1 preparation, and the prescribing information provided by dozens of manufacturers in China. The same composition was also reported in the instructions for the drug produced by Brazil TRB Pharmaceutical Company. In addition, only one component described as GM1(d18:1/C18:0) has been identified by TRB pharmaceutical company in Argentina. These structural annotations are inconsistent with the description of research literature for human, bovine and murine brain extracts that mainly contain d18:1 and d20:1 sphingosine. It is very challenging to characterize GM1 regioisomers from microheterogeneous natural sources. However, using MOCECA-synthesized GM1 analogs, we were able to resolve the contradicting descriptions of GM1 components in different fields.

The composition of commercially available GM1 drugs sourced from multiple Asian pharmaceutical companies such as QILU pharmaceutical, Science Sun pharmaceutical, and HAYIDA pharmaceutical and sourced from South American Brazilian

and Argentine TRB pharmaceutical companies were all identified. We provide representative data to identify commercialized GM1 drugs Sygen™ sold by Qilu Pharmaceutical as an example. Ultra-high-performance liquid chromatography (UHPLC) and tandem mass spectrometry (MS/MS) were used as the first method to analyze MOCECA-synthesized GM1 (d20:1/C18:0), GM1 (d18:1/C20:0), and a commercially available GM1 named Sygen™. GM1 (d18:1/C18:0) containing d18:1 sphingosine and C18:0 fatty acid is commonly recognized in natural extracts or commercial drugs in different fields. We observed that GM1 (d18:1/C18:0) in Sygen™ had an absorption peak at 4.00 min (Fig. 5a, iii) and an abundant ion peak at m/z 564 caused by the cleavage of glycosidic bond. The corresponding intact ganglioside fragment was located at m/z 1544 (Fig. 5b, iv). Next, we determined the structure of the second component in Sygen™. GM1 (d20:1/C18:0), GM1 (d18:1/C20:0), and the second component in Sygen™ had absorption peaks at 4.19–4.20 min (Fig. 5a) and could not be distinguished by retention time. However, under optimized conditions, different isomeric structures displayed unique fragmentation patterns in negative-ion electrospray MS/MS. As shown in Fig. 5b (i) and (iii), the abundant ion peaks of GM1 (d20:1/C18:0) and isomers in Sygen™ at m/z 308 and 592 were caused by the cleavage of amide and glycoside bonds, respectively. The corresponding intact gangliosides fragments were located at m/z 1572, and these fragments showed the same structure in MS/MS. As shown in Fig. 5b (ii), the abundant ion peaks of isomer GM1 (d18:1/C20:0) at m/z 336 and 592 were caused by the cleavage of amide and glycoside bonds, respectively. The difference of fatty aldehyde information fragment peaks caused by amide bond breaking

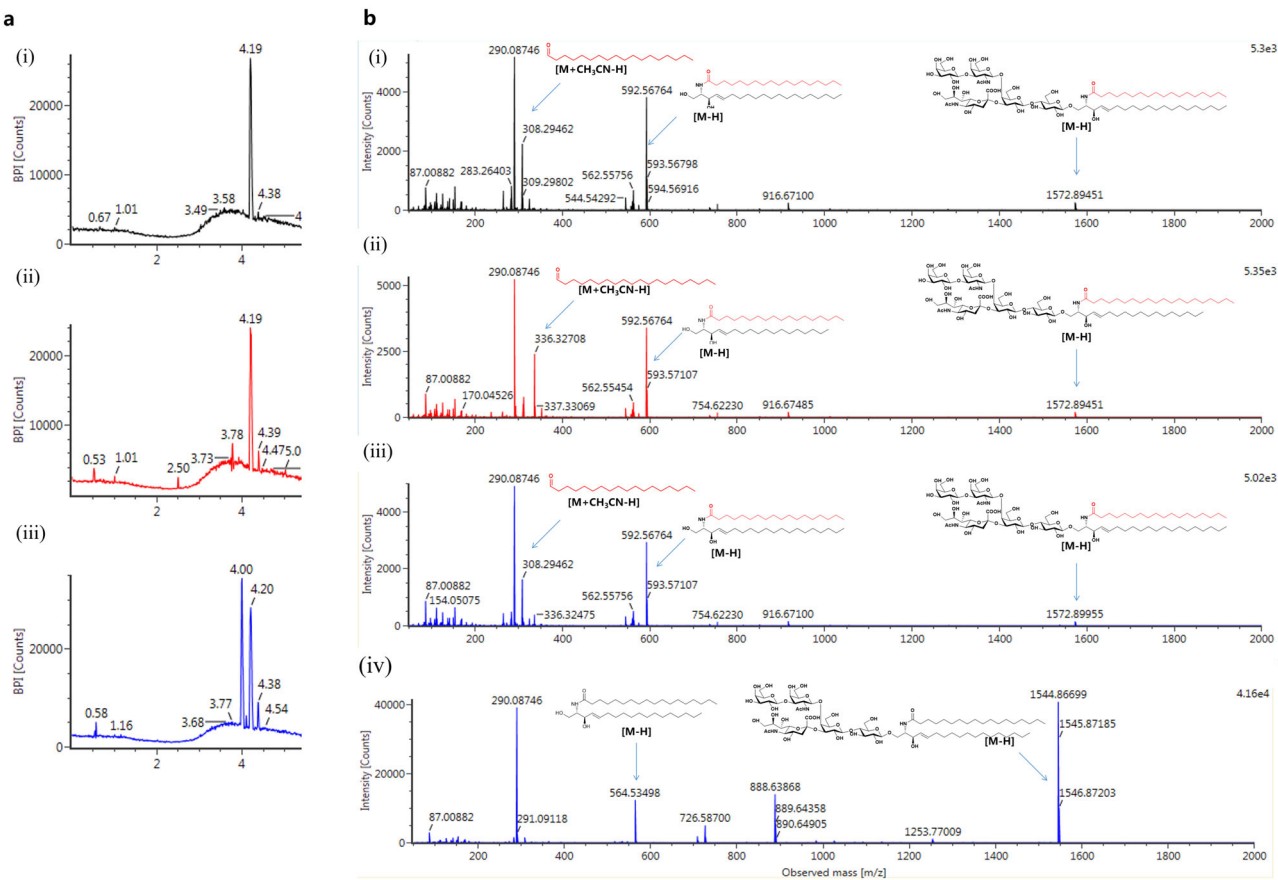

**Fig. 5 UHPLC-MS/MS of MOCECA-synthesized GM1 analogs and commercially available GM1. a** UHPLC spectra of (i) GM1 (d18:1/C18:0), (ii) GM1 (d20:1/C18:0), and (iii) Sygen™. **b** MS/MS spectra of (i) GM1 (d20:1/C18:0), (ii) GM1 (d18:1/C20:0), (iii) Sygen™ isomers, and (iv) Sygen™ GM1 (d18:1/C18:0).

confirmed that the main components of Sygen™ are GM1 (d18:1/C18:0) and GM1 (d20:1/C18:0), which have different sphingosines, and not GM1 (d18:1/C18:0) and GM1 (d18:1/C20:0), which are widely considered to be the components contained different fatty acids in the pharmaceutical field. Differences in informative fragment peaks detected by UHPLC-MS/MS provided a simple approach to identify the components of a commercial GM1 drug, and this method can also be applied for sequencing of other glycosphingolipids without derivatization.

In addition to distinguishing isomers by characterizing differences in fatty acid chains, we verified the exact structures of the two components of commercially available GM1 by identifying the glycosphingosine products of MOCECA-synthesized analogs and Sygen™ hydrolyzed by the SA_SCD deacylation system. HPLC combined with HRMS revealed that the commercial GM1 drug has two components (Fig. 6a, part IV in SI.). The effective SA_SCD hydrolysis system developed in our previous study can remove the fatty acids from GM1, producing GM1βSph with a yield of more than 95%[48]. Thus, we can accurately identify the composition of commercial GM1 by distinguishing the types of glycosylsphingosine. With MOCECA-synthesized GM1 analogs in hand, we prepared two GM1 mixtures, d18:1/C18:0-d20:1/C18:0 and d18:1/C18:0-d18:1/C20:0 ceramides, for hydrolysis using the enzymatic deacylation system. The purified hydrolysates of GM1 mixtures including the d18:1/C18:0 and d20:1/C18:0 ceramides showed two different absorption peaks and accurate molecular weights of GM1βSph (d18:1) and GM1βSph (d20:1) (part IV in SI); these results were consistent with those for the hydrolysates of Sygen™ (Fig.6b, d and part IV in SI). However, the purified hydrolysate of GM1 mixtures including the d18:1/C18:0 and d18:1/C20:0 ceramides only showed one absorption peak and one accurate molecular weight (part IV in SI); this peak corresponded with the GM1βSph (d18:1) intermediate synthesized using the MOCECA strategy (Fig. 6c). Based on the current description of commercial GM1 (with C18:0 and C20:0 saturated fatty acids) in the pharmaceutical field, the purified hydrolysate obtained by the SA_SCD hydrolysis system should be a single structure, which is inconsistent with the composition of Sygen™ actually observed. Our results verified the conclusion that commercially available GM1 drugs consist of two components with d18:1 and d20:1 sphingosine instead of C18:0 and C20:0 fatty acids. Using the MOCECA strategy, we achieved the rapid and accurate identification of the exact structures of GM1 in commercial drugs.

## Conclusion
As the important components on the membranes of vertebrate cells with great diversity, gangliosides possess tremendous potential biological functions. However, there is still a lack of a flexible and effective strategy to systematically produce ganglioside analogs with various modifications, which poses a great challenge to achieving accurate and high-yielding synthesis of target structures. In order to obtain structurally well-defined ganglioside analogs for thoroughly understanding their structure-activity relationships and developing new drugs, we design the MOCECA strategy to achieve large-scale and precisely controlled biosynthesis of single ganglioside analogs. Four improved catalytic modules characterized by simplicity, high yield, and good compatibility were successfully combined to form this customized biocatalysis strategy. Five gangliosides with therapeutic promising and ten GM1 analogs with diverse sphingosines and fatty acids were synthesized typically to demonstrate the effectiveness and universality of the MOCECA strategy to access ganglioside analogs. The MOCECA strategy is convenient for process amplification. Taking GM1 (d18:1/C18:0) as an example, we used this strategy to carry out the industrial production of ganglioside in the form of modular assembly, achieving hectogram scale preparation for the first time. It is a promising approach to provide designated glycolipid derivatives for scientific research.

The composition of various sphingosines or fatty acids in mammalian GM1 changes with age, and the type of fatty acid has been shown to affect the biological functions of GM1[21,29], which may be related to the development and aging of the nervous system. However, the structure-activity relationship of single GM1 analogs in the nervous system has not been carefully studied. To address this problem, various MOCECA-synthesized GM1 analogs with different ceramides were evaluated for neurobiological activity. Six major saturated analogs in the human brain were synthesized, including two core medicinal components with d18:1/C18:0 and d20:1/C18:0 structures. Four unsaturated fatty acid analogs were also assessed. The results of neurobiological activity assays showed that the structures of the sphingosine or fatty acid modules in GM1 can influence its effects on neurite outgrowth and cell viability. This indicates that the microheterogeneity of extracted GM1 needs to be considered in drug production. The customized biosynthesis of GM1 analogs with specific modifications by MOCECA strategy is useful for exploring their pharmacological effects on central nervous system disease.

By consulting the literature and drug description information, we found that there is a difference in the main composition of GM1 reported in drug package inserts and natural components reported in the academic literature. To resolve this contradiction, MOCECA-synthesized GM1 and GM1βSph analogs were used to identify the components of commercialized GM1 drugs named Sygen™. By analyzing the fatty aldehyde information fragment peak from UHPLC-MS/MS and the types of glycosphingosines present in GM1 hydrolysates, Sygen™ was confirmed to consist of two main GM1 components, one with d18:1 sphingosine and one with d20:1 sphingosine, rather than annotated as a two-component mixture consist of C18:0 and C20:0 fatty acids in various pharmaceutical documents including drug standard formulated by CFDA, GM1 patent application and drug package inserts. This suggests that the microheterogeneity of commercially available GM1 drugs has not been effectively characterized in the pharmaceutical field. It means that the neurobiological functions of specific structures in medicinal mixtures have not been clearly explained, which may bring potential risks for drug application in terms of safety and indication. This issue should be of wide concern in the fields of medicine and pharmacy.

In summary, we use the MOCECA strategy to achieve controlled and large-scale biosynthesis of ganglioside analogs presenting various glycan and ceramide epitopes. Based on the analogs synthesized using this strategy, we showed that unique ceramide modifications can influence the neurological activities of GM1 components, and we revealed the exact two-component structures of commercially available GM1 drug. The MOCECA strategy not only can be used for the preparation of ganglioside analogs, but also can provide new insight into the catalytic synthesis of other GSL compounds in vitro. We have initiated research aimed at obtaining various GSLs and derivative libraries. It is anticipated that this technology can be used for exploring the regulatory effects of GSL analogs in physiological and pathological processes of life, which has important biological significance.

## Methods
**General procedure for the identification of glycosylsphingosine analogs by HPLC.** An analytical Welch ultimate XB-C18 column (3.5 μm, 4.6 mm × 250 mm) was used to detect the contents of glycosylsphingosines and for final purity analysis. The signal was detected by an evaporative light-scattering detector. The analysis

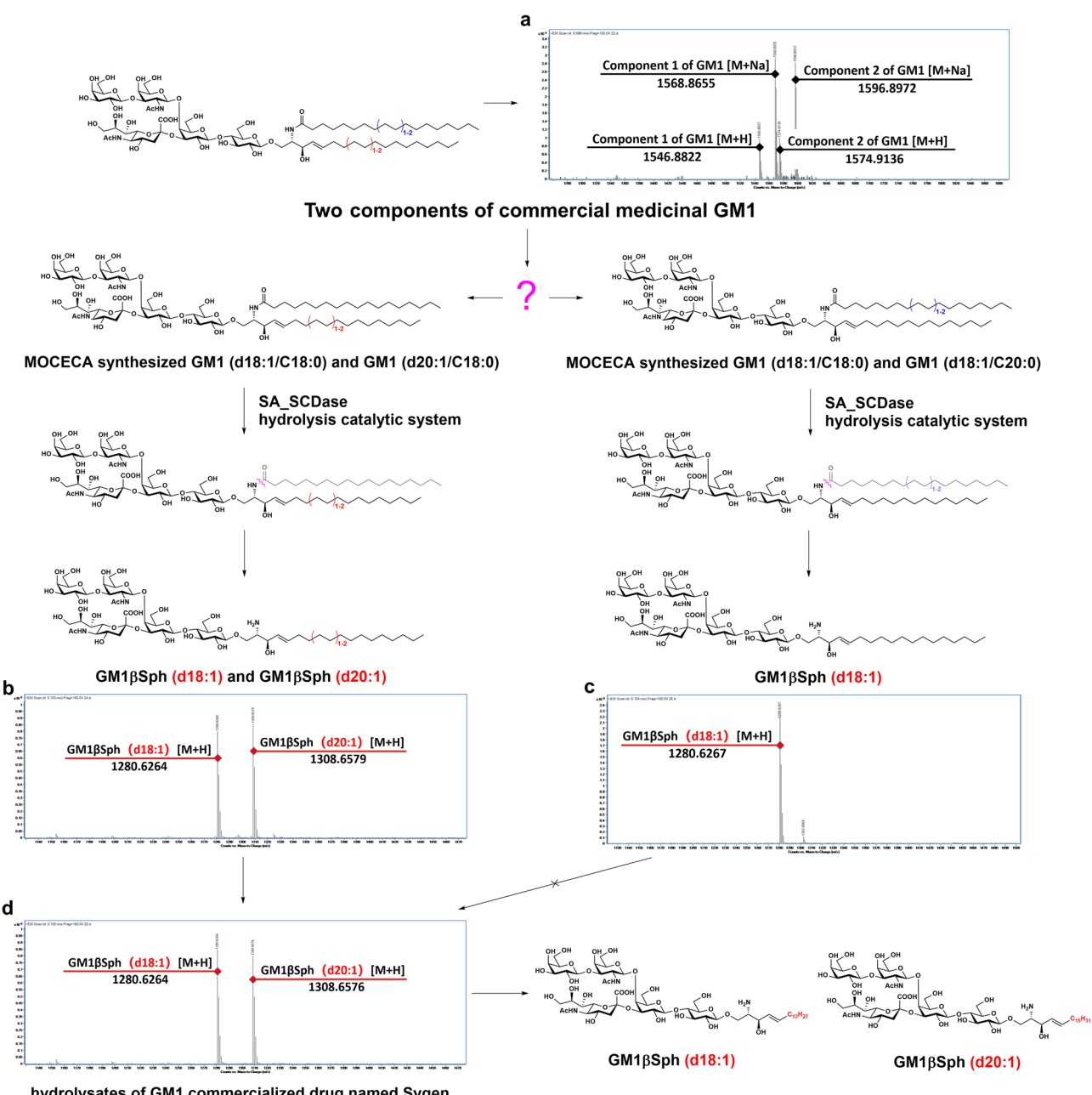

**Verification using MOCECA-synthesized GM1 and GM1βSph derivatives**

**Fig. 6 Identification of the composition of commercially available medicinal grade GM1 using the MOCECA strategy. a** HRMS spectrum for a commercial GM1 drug named Sygen™. **b** HRMS spectrum for glycosphingosine products of MOCECA-synthesized GM1 analogs mixed with GM1 (d18:1/C18:0) and GM1 (d20:1/C18:0) hydrolyzed by SA_SCD. **c** HRMS spectrum for glycosphingosine products of MOCECA-synthesized GM1 analogs mixed with GM1 (d18:1/C18:0) and GM1 (d18:1/C20:0) hydrolyzed by SA_SCD. **d** HRMS spectrum for glycosphingosine products of Sygen™ hydrolyzed by SA_SCD.

was performed under the following running conditions: 0–20 min, 50% A and 50% B; flow rate: 1 mL/min; injection volume: 20 µL; column temperature: 40 °C. Phase A: acetonitrile. Phase B: 0.1% formic acid.

**General procedure for the identification of gangliosides and GM1 analogs by HPLC.** An analytical Waters Sunfire C18 color column (3.5 µm, 4.6 mm × 150 mm) was used to monitor the formation and purity of GM1 analogs. The signal was detected by a UV detector at 205 nm. The analysis was performed under the

following gradient running conditions: 0–60 min, 100–55% B; 60–61 min, 55–100% B; 61–70 min, 100% B; flow rate: 1 mL/min; injection volume: 20 µL; column temperature: 40 °C. Phase A: acetonitrile. Phase B: 30% $H_2O$ with 0.01 mol $L^{-1}$ potassium dihydrogen phosphate/70% acetonitrile (v/v).

**Assessment of neurite outgrowth activities of MOCECA-synthesized GM1 and GM1βSph analogs in Neuro2a cells.** Neuro2a cells were seeded in 24-well plates with 10% fusion and cultured for 24 h in DMEM supplemented with 10% FBS. The

next day, GM1 and GM1βSph analogs were added separately to the cells at a concentration of 25 µM, and cells were cultured for another 24 h. The morphological changes of Neuro2a cells were observed, and cells were photographed under a phase-contrast microscope. At least 300 cells were counted in each well. The length of the longest neurite and the proportion of cells bearing a neurite were recorded for more than 30 cells with processes. Significant differences among groups were calculated using Newman-Keuls multiple comparison test following one-way ANOVA analysis in GraphPad Prism software (*$p < 0.05$, **$p < 0.01$, ***$p < 0.001$).

**Procedure for UHPLC-MS/MS detection of MOCECA-synthesized analogs and commercialized GM1.** A BEH C18 color column (1.7 µm, 2.1 mm × 100 mm) was used to detect MOCECA-synthesized analogs and Sygen™. The signal was detected by a UV detector at 205 nm. The analysis was performed under the following gradient running conditions: 0–2 min, 5%–50% B; 2–4 min, 50%–80% B; 4–7 min, 80%–100% B; 7–9.5 min, 100% B; 9.5–9.6 min, 100–5% B; 9.6–12 min, 5% B; flow rate: 0.4 mL/min; injection volume: 20 µL; column temperature: 45 ℃. Phase A: 0.1% formic acid in water (v/v); Phase B: 0.1% formic acid in acetonitrile (v/v). MS/MS conditions were as follows: Acquisition mode: MS$^E$; Ionization mode: ESI negative; Capillary voltage: 2 KV (negative); Cone voltage: 40 V; Desolvation temp.: 450 ℃; Source temp.: 115 ℃; Collision energy: 6 eV/20–45 eV.

**Reporting summary.** Further information on research design is available in the Nature Portfolio Reporting Summary linked to this article.

## Data availability

All data needed to support the conclusion are present in the article, supplementary information and supplementary data. Supplementary information comprehensively provide the experimental procedures involved in various catalytic modules of the MOCECA strategy including chemical synthesis, streamlined multienzyme glycosylation, enzymatic transglycosylation, enzymatic acylation. The experimental procedures for component identification of GM1 commercialized drugs are also described. All data and spectra of HRMS, and HPLC for compounds in the article are listed in supplementary information. All NMR spectra for compounds in the article are listed in supplementary data.

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

## Acknowledgements

This study was financially supported by the National Key R&D Program of China (2018YFE0200501, 2020YFA0907900), National Natural Science Foundation of China (Grant Nos. 32030063 and 21877063), National Key R & D Plan "Blue Granary Science and Technology Innovation" (2019YFD0901902), and Shanghai Pilot Program for Basic Research - Shanghai Jiao Tong University. H. Cheng gratefully acknowledges the Guangdong Basic and Applied Basic Research Foundation (2021A1515111039) and Shenzhen Science and Technology Program (No. JCYJ20220530113815035).

## Author contributions

X.J. performed most experiments; H.C. performed the analysis of NMR spectra for compounds; X.H.C. designed cell experiments; X.F.C designed the sphingosine synthesis experiment; C.X., F.D., and H.Q. designed the enzymatic glycosylation experiment; G.Y., Y.F., and P.W. designed the project and wrote the manuscript. X.J., H.C., and G.Y. revised the manuscript. All authors have given approval to the final version of the manuscript.

## Competing interests

The authors declare no competing interests.
