## [Peer Review File · Communications Chemistry]

Reviewers' comments:

Reviewer #1 (Remarks to the Author):

The paper by Xuefeng Jin and colleagues describes a modular chemo-enzymatic cascade assembly (MOCECA) strategy for the medium to large-scale synthesis of gangliosides with different glycan and lipid structures. The modularity of this strategy makes it suitable for the production of gangliosides with different sugar and ceramide characteristics, which would allow for an in-depth characterization of the pharmacological and biological properties of different ganglioside structures. As a proof of principle, the authors claim that GM1 moieties with different ceramide composition have different effects on neurite outgrowth and cell viability in neuroblastoma N2a cells, and different toxicity profile, highlighting the importance of defining the structure-activity relationship for gangliosides from different sources or isolated from natural sources as mixed species. In addition, the authors define the composition of ganglioside GM1 present in Sygen, the commercial name for ganglioside purified from porcine brain.

Strengths

The work by Xuefeng et al. provides an important strategy for large scale preparation of gangliosides with different structures. This is not a trivial task, as large scale synthesis of gangliosides with different ceramide structures is notoriously difficult. Furthermore, the MOCECA strategy designed by the authors is timely in light of the renewed interest in the therapeutic potential of gangliosides. The preparation of D-sphingosines of different lengths is particularly important to recapitulate the mixed structure of brain gangliosides, and to allow for the preclinical experimental testing of the biological and pharmacological activity of ganglioside species with precise ceramide composition.

Weaknesses

A major weakness of the manuscript is represented by the cell experiments, as detailed below, and I cannot help wondering whether these are necessary at all for the scope of this work.

Details and additional weaknesses are listed below.

- 1) The experiments that address the effects of ganglioside GM1 with different composition of the ceramide moiety are not compelling and the results might be overstated and misinterpreted, since the MTT assay does not specifically measure cell death, and is affected by changes in cell proliferation, cell adhesion and mitochondrial metabolism that could potentially be induced by the individual gangliosides and confound results. At concentrations above 75-100 μ M, certain gangliosides can decrease cell adhesion and cause the cells to lift from the plate, which could be mistaken for cell death.
- 2) The use of t-test for the analysis of the neurite extension data is not appropriate and might be misleading.
- 3) The manuscript would benefit from professional editing to clarify many statements and sentences throughout the manuscript. As an example, lines 31-33, 38-39, 49-50, line 56 (aggregation and precipitation of GM2??? To indicate presumably lysosomal accumulation?). Line 120: What do the authors mean with "de value"?

4) The wording “ganglioside derivatives” is often used to indicate different ganglioside species, and it is misleading, as it suggests the gangliosides are chemically modified, when instead gangliosides with different sphingosine and fatty acids are just different species of the same ganglioside.

5) Figure legends are too small and often not easy to read.

6) Some of the references cited are not appropriate or do not specifically support the statements mentioned, for example, reference 3,4, line 50, reference 145, line 55. References 21, 27, 49 describe changes in the ceramide species of gangliosides with ageing, which is not necessarily associated with neurodegeneration, therefore, the statement on line 273 should be changed to “involved in brain development and ageing”.

7) It is stated that three GM1 species bearing d20:1 sphingosine promote neurite outgrowth as shown in Fig. 4b and c, but frankly I struggled identifying which one are these compounds from the graph.

8) The statement on lines 383-384 that commercially available GM1 is widely used for the treatment of central nervous system injury and Parkinson’s disease should be eliminated or changed. In North America, Australia and Europe, GM1 is not used for the treatment of patients. If clinical use is made elsewhere in the world, this should be specified as not to mislead the reader. Similarly, in the introduction, the authors state that Sygen is the most widely used kind of gangliosides in the clinic. This statement requires a more specific description of where and when Sygen was used, as I am not aware that Sygen is currently used in Europe or North America at least.

9) The authors keep referring to “commercialized GM1” without specifying who sells the product, and whether the authors are referring to only one commercial source or multiple. The entire section relative to commercialized GM1 is confusing. Importantly, the source of Sygen for this study is not mentioned anywhere.

10) Lines 335-337: The authors state that the observation that d20:1 species improve cell viability, combined with the knowledge that the proportion of GM1 component containing D20:1 sphingosine in the brain increases with age, suggests that these “these derivatives may be associated with the development of the central nervous system”. This conclusion seems quite arbitrary and should be eliminated.

Reviewer #2 (Remarks to the Author):

Gangliosides play an important part in biological regulatory roles in multiple major human diseases, such as nervous system diseases, lysosomal storage diseases, immune system diseases, and cancers, and received extensive attention in developing targeted drugs and diagnostic markers. However, the difficulty in scalable preparation prevents exploring structure-activity relationships and identifying drug ingredients. Dr. Yang group reported a highly modular chemo-enzymatic cascade assembly (MOCECA) strategy for customized and large-scale synthesis of ganglioside derivatives with various glycan and

ceramide epitopes. This method could access several gangliosides with therapeutic promising and systematically prepared primary GM1 derivatives with diverse ceramides found in human brain. They also developed industrial production of ganglioside GM1 in the form of modular assembly at hectogram scale. Based on the derivatives synthesized using this strategy, they showed that unique ceramide modifications can influence the neurological activities of GM1 components and revealed the exact two-component structures of commercially available GM1 drug. This method has potential use for exploring the regulatory effects of GSL derivatives in physiological and pathological processes of life. Therefore, I recommend the work to be published in Communications Chemistry if the following points could be addressed.

1. The resolution of Fig 1, Fig 3, Fig 6 should be improved.
2. In Fig 2, the length of arrow should be adjusted; The space between the reagent and arrow should be consistent; the structure of the organic compound should be clean up.
3. The amount of the reagents should be noted below Fig 2, for example, showing the equiv. or mol% for the reagents.
4. In SI, in Experimental procedures 1, 1. Scalable and cost-effective preparation of D-sphingosines with high purity by chemical synthesis, the mole, equiv. or mol% should be noted.
5. If the product has been reported in previous publications, please cite the paper and show if the results agree with the reported data.

Reviewer #3 (Remarks to the Author):

The authors presented a modular chemoenzymatic cascade approach for the production of a range of gangliosides that can be scaled to hectogram isolated product. Gangliosides are glycosphingolipids that mainly occur in the nervous system as cell membrane components, and play a role in diverse biological functions. A number of ganglioside derivatives have been tested for their neurobiological activity using an in vitro model. Furthermore, a commercial GM1 preparation was analyzed for their fatty acid composition using the newly synthesized GM1 derivatives as reference compounds. The new chemosynthetic approach to produce a wide range of gangliosides is important and suitable for CommsChem after revisions. The manuscript (incl. SI) is well written and the cascade is extensively described, but a number of points can be improved.

- 1) add the amounts (or concentrations) of enzyme used in the reactions (now they are only in the SI). Can the enzymes be re-used. Were the enzymes immobilized? It is hard to estimate the cost of the overall process if the cost and effort of enzyme production is not taken into account.
- 2) A large number of enzymes is used in the cascade. How were the enzymes purified, or were cell extracts used? How were the enzymes obtained. In the SI only references to papers were included, it would be helpful to add a brief description of the expression and purification so the reader does not have to scan the methods of 12 papers to recreate the procedures.
- 3) indicate in the cascade where the intermediate products have to be purified (e.g. in fig 1).
- 4) some of the text and structures in the figures are very small. Please follow journal guidelines and optimize visibility of all figure elements. For example Fig 6.
- 5) the "industrial and cost-effective preparation of GM1 derivatives" used gram-scale enzyme amounts

to produce 100 gram-scale product in a 10 L reactor. The authors should add some cost estimates to show the cost-effectiveness of their process. In my opinion this is a very nice proof of concept, but I am not confident about the overall costs of the process.

6) The biological tests mainly serve to show (general) biological activity of the products, which is fine. As a study on the neurobiological activity of each product a more extensive study would be warranted.

7) indicate under "methods" what is given in the SI. Or give a more detailed description of the contents of the SI under "additional information"

8) line 394 replace "regiosomer" by "regioisomer"

Reviewer #4 (Remarks to the Author):

Guangyu and coworkers report the synthesis of various ganglioside derivatives utilizing a chemoenzymatic cascade to assemble the individual fragments while employing a chemical synthetic approach towards making the ceramide component. The authors proceed to evaluate the neurobiological activities of these derivatives. This reviewer will not comment on this component of the manuscript and instead will focus on the synthesis. There are two notable synthetic advances here. One being the improved chemical synthesis of the ceramide component - previous work here has relied on particularly lengthy routes and ones that are far more expensive to carry through. The chemistry here is highly scalable and amenable to cost-effective synthesis of ceramide analogues. The second is the highly scalable nature of chemo-enzymatic cascade reactions. While the use of enzymes for assembling gangliosides have been known for a few decades now, these are rarely performed on such large scales. The authors make this point in their introduction without referencing any literature - please address this (line 83). These advances in themselves render this manuscript worthy of publication in Communications Chemistry.

Minor concerns:

The supporting information is to a high-standard. Some small points here: Did the authors check for racemization at some point after forming the alpha-amino ketone intermediate? These often have issues with racemization and would require ee% measurements to confirm enantiopurity. How did the authors confirm the relative stereochemistry resulting from the glycosylation steps?

Point-by-point responses to the Reviewer's comments

Reviewer #1 (Remarks to the Author):

The paper by Xuefeng Jin and colleagues describes a modular chemo-enzymatic cascade assembly (MOCECA) strategy for the medium to large-scale synthesis of gangliosides with different glycan and lipid structures. The modularity of this strategy makes it suitable for the production of gangliosides with different sugar and ceramide characteristics, which would allow for an in-depth characterization of the pharmacological and biological properties of different ganglioside structures. As a proof of principle, the authors claim that GM1 moieties with different ceramide composition have different effects on neurite outgrowth and cell viability in neuroblastoma N2a cells, and different toxicity profile, highlighting the importance of defining the structure-activity relationship for gangliosides from different sources or isolated from natural sources as mixed species. In addition, the authors define the composition of ganglioside GM1 present in Sygen, the commercial name for ganglioside purified from porcine brain.

Strengths

The work by Xuefeng et al. provides an important strategy for large scale preparation of gangliosides with different structures. This is not a trivial task, as large scale synthesis of gangliosides with different ceramide structures is notoriously difficult. Furthermore, the MOCECA strategy designed by the authors is timely in light of the renewed interest in the therapeutic potential of gangliosides. The preparation of D-sphingosines of different lengths is particularly important to recapitulate the mixed structure of brain gangliosides, and to allow for the preclinical experimental testing of the biological and pharmacological activity of ganglioside species with precise ceramide composition.

Weaknesses

A major weakness of the manuscript is represented by the cell experiments, as detailed below, and I cannot help wondering whether these are necessary at all for the scope of this work.

Details and additional weaknesses are listed below.

Responses to Reviewer #1:

Point #1: *The experiments that address the effects of ganglioside GM1 with different composition of the ceramide moiety are not compelling and the results might be overstated and misinterpreted, since the MTT assay does not specifically measure cell death, and is affected by changes in cell proliferation, cell adhesion and mitochondrial metabolism that could potentially be induced by the individual gangliosides and confound results. At concentrations above 75-100 μ M, certain gangliosides can decrease cell adhesion and cause the cells to lift from the plate, which could be mistaken for cell death.*

Response: We thank the Reviewer for the helpful guidance about improving our study. In this study, our focus is to describe a highly modular chemo-enzymatic cascade assembly (MOCECA) strategy for customized and large-scale synthesis of ganglioside derivatives with various glycan and ceramide epitopes. The preliminary exploration for the biological activity of various GM1 analogues is only an expanded experimental observation based on synthetic compounds. As Reviewer stated, MTT assay cannot specifically measure the cytotoxicity of GM1 compounds on Neuro2a cells. Moreover, it is not necessary for the theme of this work. Therefore, we have removed this section to ensure the results of the manuscript is rigorous.

Point #2: *The use of t-test for the analysis of the neurite extension data is not appropriate and might be misleading.*

Response: We would like to thank the Reviewer for this helpful comment. Due to the fact that the analysis of the neurite extension data is the comparison between multiple groups of Neuro2a cells treated with different GM1 species, t-test is not suitable and one-way ANOVA should be used. After reanalyzing and proofreading the results of neurite outgrowth, we used one-way ANOVA to recalculate the data and made revisions to the manuscript to present the correct statistical results, which were shown in Figure 4b and 4c. The revised experimental results still support the overall

conclusion of the manuscript, namely GM1 analogues with diverse ceramide structures have different neurobiological activities in promoting neurite outgrowth.

Point #3: *The manuscript would benefit from professional editing to clarify many statements and sentences throughout the manuscript. As an example, lines 31-33, 38-39, 49-50, line 56 (aggregation and precipitation of GM2??? To indicate presumably lysosomal accumulation?). Line 120: What do the authors mean with “de value”?*

Response: We thank the Reviewer for the helpful guidance.

Firstly, we have made revisions to the sentences in lines 31-33, 38-39, 49-50 to make the original text more clear.

Secondly, we have rephrased the content in line 56. The updated version reads as follows: The aggregation and precipitation of GM2 in lysosome of special brain regions is the typical pathological feature of Tay Sachs, a sphingolipid lysosomal storage disorder.

Thirdly, please allow us to explain enantiomers and diastereomers before “de value”. In chemistry, an enantiomer is one of two stereoisomers that are mirror images of each other but cannot be superposed. Diastereomers are stereoisomers that are not enantiomers and are not associated as object and mirror image. Diastereomers are not mirror images of each other and are non-superimposable. The diastereomers occur when two or more stereoisomers of a compound have a different configuration located at one or more stereocenters. “de” is the abbreviation of diastereomeric excess (Pages 133-193 in the book Chapter 4 of March’s Advanced Organic Chemistry, 8th edition). The diastereomeric excess helps in quantifying the stereoselectivity. It measures the degree to which a sample consists of one diastereomer in greater amounts as compared to the other. The following is the formulae used for the diastereomeric excess calculation, where [D1] and [D2] are two diastereomeric.

$$\text{de}\% = \frac{|D1 - D2|}{D1 + D2} \times 100\%$$

When there is a 1:1 mixture of two diastereomers, the value of de = 0%, whereas for the value of the diastereomeric pure compound, the value of de = 100%.

Point #4: The wording “ganglioside derivatives” is often used to indicate different ganglioside species, and it is misleading, as it suggests the gangliosides are chemically modified, when instead gangliosides with different sphingosine and fatty acids are just different species of the same ganglioside.

Response: Many thanks for the Reviewer’s suggestion. We think “ganglioside analogues” are more accurate than “ganglioside derivatives” in this situation. Ganglioside analogues are used instead in the manuscript.

Point #5: Figure legends are too small and often not easy to read.

Response: We thank the Reviewer for this helpful comment. We have enlarged the font size of the figure legends to enhance readability.

Point #6: Some of the references cited are not appropriate or do not specifically support the statements mentioned, for example, reference 3,4, line 50, reference 14, line 55. References 21, 27, 49 describe changes in the ceramide species of gangliosides with ageing, which is not necessarily associated with neurodegeneration, therefore, the statement on line 273 should be changed to “involved in brain development and ageing”.

Response: We thank the Reviewer for pointing this out. As Reference 3 and 4 may be not appropriate in the manuscript, we have revised the references to specifically support the statements mentioned. The updated version of Reference 3 and 4 reads as follows:

[3] Chowdhury, S. & Ledeen, R. The Key Role of GM1 Ganglioside in Parkinson's Disease. *Biomolecules* 12(2022).

[4] Maglione, V. et al. Impaired ganglioside metabolism in Huntington's disease and neuroprotective role of GM1. *J Neurosci* 30, 4072-80 (2010)

Although GM1 application as a therapeutic drug in neurological diseases including Parkinson's disease, Alzheimer's disease, and spinal cord injury has been described in section 10 of the original reference 14, it may not be appropriate enough. Therefore, we

have revised the reference to better support the content in line 55.

The updated version of Reference 14 reads as follows:

[14] Magistretti, P.J. et al. Gangliosides: Treatment Avenues in Neurodegenerative Disease. *Front Neurol* 10, 859 (2019).

As “ageing” is more accurate and suitable compared to “degeneration”, we have made this change in line 273 according to your suggestion.

Point #7: It is stated that three GM1 species bearing d20:1 sphingosine promote neurite outgrowth as shown in Fig. 4b and c, but frankly I struggled identifying which one are these compounds from the graph.

Response: We thank the reviewer for the helpful guidance. In order to facilitate readers to more intuitively distinguish the neurobiological activities of various GM1 analogues, we added the annotation rules for different types of GM1 analogues in legends of Figures 4 as follows:

“GM1 with d18:1 sphingosine were shown in blue or dark blue. GM1 with d20:1 sphingosine were shown in pink or red. GM1 β Sph analogues were shown by the bar with horizontal lines. Saturated GM1 analogues were labeled by the chart of hollow bar. Unsaturated GM1 analogues are labeled by the bar chart with diagonal line.”

In order of six GM1 saturated species, four GM1 with unsaturated fatty acid species and two GM1 β Sph intermediates, we analyzed their neurite growth promoting activities. Compared to the untreated group (CON), three GM1 species bearing d20:1 sphingosine (the fifth, seventh, and ninth bar) can promote neurite outgrowth with statistical differences. Among three GM1 species bearing d18:1 sphingosine, GM1 (d18:1/C16:0) and GM1 (d18:1/C20:0) showed relatively weak activities, and there was no statistical difference between these treatment groups and the control group. GM1 (d18:1/C18:0), had an effect on neurite outgrowth similar to that of d20:1 sphingosine. Moreover, we further analyzed the neurobiological activities of four GM1 unsaturated species and two GM1 β Sph intermediates. GM1 (d18:1/C16:1), GM1 (d18:1/C18:1), GM1 (d20:1/C16:1), and GM1 (d20:1/C18:1) groups all showed stronger activities in promoting neurite growth compared with untreated group. In addition, GM1 β Sph

(d18:1) and GM1 β Sph (d20:1) also increased neurite outgrowth and length.

Therefore, compared to the untreated group, ten GM1 species and two GM1 β Sph intermediates have been shown to promote neurite outgrowth in to varying degrees. This is stated in line 371: “Taken together, our assays showed that GM1 analogues with diverse ceramide structures have different promoting activities on neurite outgrowth in cells.”

Point #8: The statement on lines 383-384 that commercially available GM1 is widely used for the treatment of central nervous system injury and Parkinson’s disease should be eliminated or changed. In North America, Australia and Europe, GM1 is not used for the treatment of patients. If clinical use is made elsewhere in the world, this should be specified as not to mislead the reader. Similarly, in the introduction, the authors state that Sygen is the most widely used kind of gangliosides in the clinic. This statement requires a more specific description of where and when Sygen was used, as I am not aware that Sygen is currently used in Europe or North America at least.

Response: We thank the Reviewer for raising this point. Commercialized GM1 drug is produced and is widely used to treat neurological diseases in China, Brazil, and Argentina. According to your suggestion, we eliminated the statement on lines 383-384 that commercially available GM1 is widely used for the treatment of central nervous system injury and Parkinson’s disease, which make the expression of the manuscript more rigorous. Ganglioside GM1 was reported to be applied in numerous previous clinical and preclinical studies for various neurological diseases, including spinal cord injury, stroke, Parkinson’s disease, Huntington’s disease, Alzheimer’s disease¹. Through updating the content in the second and fourth paragraph of the introduction, we provide a more specific description of where and when Sygen was used.

Reference

[1] Magistretti, P.J. et al. Gangliosides: Treatment Avenues in Neurodegenerative Disease. *Front Neurol* 10, 859 (2019)

Point #9: The authors keep referring to “commercialized GM1” without specifying who sells the product, and whether the authors are referring to only one commercial source or multiple. The entire section relative to commercialized GM1 is confusing. Importantly, the source of Sygen for this study is not mentioned anywhere.

Response: We thank the Reviewer for this helpful guidance. We have conducted drug components identification for “commercialized GM1” produced by multiple Asian pharmaceutical companies including Qilu pharmaceutical, Science Sun pharmaceutical, HAYIDA pharmaceutical, as well as produced by South American Brazilian and Argentine TRB pharmaceutical companies. All commercialized GM1 drug sourced from these above-mentioned companies were confirmed as a mixture containing two components with d18:1 and d20:1 sphingosine instead of C18:0 and C20:0 fatty acids. We provide representative data to identify of the components in commercialized GM1 drugs Sygen™ sold by Qilu Pharmaceutical as an example. The commercialized drugs used for components identification come from multiple pharmaceutical companies. We have added descriptions of the sources of these commercialized drugs in line 240.

Point #10: Lines 335-337: The authors state that the observation that d20:1 species improve cell viability, combined with the knowledge that the proportion of GM1 component containing D20:1 sphingosine in the brain increases with age, suggests that these “these derivatives may be associated with the development of the central nervous system”. This conclusion seems quite arbitrary and should be eliminated.

Response: We thank the Reviewer for raising this point. According to your suggestion, we eliminated the imprecise conclusion.

Reviewer #2 (Remarks to the Author):

Gangliosides play an important part in biological regulatory roles in multiple major human diseases, such as nervous system diseases, lysosomal storage diseases, immune system diseases, and cancers, and received extensive attention in developing targeted drugs and diagnostic markers. However, the difficulty in scalable preparation prevents exploring structure-activity relationships and identifying drug ingredients. Dr. Yang

group reported a highly modular chemo-enzymatic cascade assembly (MOCECA) strategy for customized and large-scale synthesis of ganglioside derivatives with various glycan and ceramide epitopes. This method could access several gangliosides with therapeutic promising and systematically prepared primary GM1 derivatives with diverse ceramides found in human brain. They also developed industrial production of ganglioside GM1 in the form of modular assembly at hectogram scale. Based on the derivatives synthesized using this strategy, they showed that unique ceramide modifications can influence the neurological activities of GM1 components and revealed the exact two-component structures of commercially available GM1 drug. This method has potential use for exploring the regulatory effects of GSL derivatives in physiological and pathological processes of life. Therefore, I recommend the work to be published in Communications Chemistry if the following points could be addressed.

Responses to Reviewer #2:

Point #1: The resolution of Fig 1, Fig 3, Fig 6 should be improved.

Response: We would like to thank the Reviewer for this helpful comment. According to your suggestion, the resolution of Fig 1, Fig 3, Fig 6 have been improved.

Point #2: In Fig 2, the length of arrow should be adjusted; The space between the reagent and arrow should be consistent; the structure of the organic compound should be clean up.

Response: We thank the Reviewer for the helpful guidance in Fig. 2 and have made revisions point by point. We have adjusted the lengths of arrows and the space between the reagent and arrow. The structures of compounds have also been cleaned up.

Point #3: The amount of the reagents should be noted below Fig 2, for example, showing the equiv. or mol% for the reagents.

Response: We thank the Reviewer for pointing this out. We have added the amount of all reagents from the starting material L-serine (200.0 g, 1.903 mol). All the changes

are revised in blue color in the legends of Fig. 2.

Point #4: In SI, in Experimental procedures 1, 1. Scalable and cost-effective preparation of D-sphingosines with high purity by chemical synthesis, the mole, equiv. or mol% should be noted.

Response: We gratefully appreciate for your valuable suggestion. We have carefully checked and added the mole amounts and equivalent numbers of chemicals used for the organic synthesis in Experimental procedures 1. All the changes are revised in blue color in supplementary information.

Point #5: If the product has been reported in previous publications, please cite the paper and show if the results agree with the reported data.

Response: We thank the Reviewer for pointing this out. In this manuscript, D-sphingosine (d18:1) has been synthesized by other methods in previous publications, and we have cited relevant literature as reference 25. We have added the description to illustrate that the experimental results are consistent with the reported data. As D-sphingosine (d20:1) has not been reported yet, we used HRMS, ^1H NMR, and ^{13}C NMR to identify the structure of the compound. Similar structures of GM3, GM2, GM1, GD3 and GD2 gangliosides with C16 saturated fatty acids have been synthesized in different ways by other publications, we have cited this literature as reference 23. Due to the differences in fatty acid chains between the gangliosides we synthesized and the compounds synthesized in other publications, we also applied HRMS, ^1H NMR, and ^{13}C NMR for structural identification of the compound.

Reviewer #3 (Remarks to the Author):

The authors presented a modular chemoenzymatic cascade approach for the production of a range of gangliosides that can be scaled to hectogram isolated product. Gangliosides are glycosphingolipids that mainly occur in the nervous system as cell membrane components, and play a role in diverse biological functions. A number of ganglioside derivatives have been tested for their neurobiological activity using an in

vitro model. Furthermore, a commercial GM1 preparation was analyzed for their fatty acid composition using the newly synthesized GM1 derivatives as reference compounds. The new chemosynthetic approach to produce a wide range of gangliosides is important and suitable for CommsChem after revisions. The manuscript (incl. SI) is well written and the cascade is extensively described, but a number of points can be improved.

Responses to Reviewer #3:

Point #1: add the amounts (or concentrations) of enzyme used in the reactions (now they are only in the SI). Can the enzymes be re-used. Were the enzymes immobilized? It is hard to estimate the cost of the overall process if the cost and effort of enzyme production is not taken into account.

Response: We would like to thank the reviewer for this helpful comment. We have added the amount of enzymes used in each enzymatic reaction in the manuscript as suggested. Most of the enzymes applied in MOCECA were observed to have good expression level in E.coli and high catalytic activity. Although the immobilization of these enzymes was not studied in this article, they have great potential to be immobilized and reuse in the catalysis. As the reviewer suggested, we have estimate the cost of the overall process for GM1 production, as you can see in the following response #5.

Point #2: A large number of enzymes is used in the cascade. How were the enzymes purified, or were cell extracts used? How were the enzymes obtained. In the SI only references to papers were included, it would be helpful to add a brief description of the expression and purification so the reader does not have to scan the methods of 12 papers to recreate the procedures.

Response: We thank the reviewer for the helpful guidance. According to your suggestion, we have provided a brief description of the expression and purification of various enzymes used in section 1 of supplementary information to facilitate readers' better understanding of this article.

Point #3: indicate in the cascade where the intermediate products have to be purified (e.g. in fig 1).

Response: Thanks for Reviewer's helpful suggestions. We have added corresponding descriptions in the legend of figure 1 to illustrate where the intermediate products have to be purified in the cascade strategy.

Added contents reads as follows:

Sphingosine derivatives produced by chemical synthesis and glycosylsphingosine derivatives synthesized by enzymatic transglycosylation needs to be purified for the next reaction (in the legend of figure 1).

Point #4: some of the text and structures in the figures are very small. Please follow journal guidelines and optimize visibility of all figure elements. For example Fig 6.

Response: We thank the Reviewer for raising this point. According to your suggestion, we have appropriately optimized the figure elements follow with journal guidelines to enhance visibility.

Point #5: the "industrial and cost-effective preparation of GM1 derivatives" used gram-scale enzyme amounts to produce 100 gram-scale product in a 10 L reactor. The authors should add some cost estimates to show the cost-effectiveness of their process. In my opinion this is a very nice proof of concept, but I am not confident about the overall costs of the process.

Response: We thank the Reviewer for the helpful guidance. According to your suggestion, we conducted a cost estimation for the preparation of 100 gram of GM1 in Supplementary Information:

Table 1. The cost accounting for preparation of 100 grams of GM1

Projects required in GM1 preparation	Cost per 100 gram	Average cost
(1) Main substrates and reagents		
Lac-F (36.42 g)	389.56 USD	3.90 USD/g
Neu5Ac (55.62 g)	77.92 USD	0.78 USD/g

CTP (89.27 g)	77.94 USD	0.78 USD/g
GalNAc (28.08 g)	150.90 USD	1.51 USD/g
ATP (151.58 g)	38.96 USD	0.39 USD/g
UTP (145.31 g)	160.70 USD	1.61 USD/g
Gal (20.98 g)	2.8 USD	0.03 USD/g
L-serine (60.42 g)	3.34 USD	0.03 USD/g
Acetyl chloride (302.11 mL)	3.13 USD	0.03 USD/g
(Boc) ₂ O (141.39 g)	2.63 USD	0.03 USD/g
Triethylamine (181.27 mL)	1.67 USD	0.02 USD/g
TBDMSCl (141.99 g)	3.63 USD	0.04 USD/g
Imidazole (64.65 g)	1.67 USD	0.02 USD/g
Dimethyl methyl phosphonate (240.18 g)	21.68 USD	0.22 USD/g
n -BuLi (679.76 mL, 2.5 M in hexane)	37.52 USD	0.38 USD/g
LiCl (36.56 g)	3.34 USD	0.03 USD/g
n -tetradecanal (93.66 g)	29.18 USD	0.29 USD/g
LiAlH(O ^t Bu) ₃ (230.21 g)	91.72 USD	0.92 USD/g
Dichloromethane (6.04 L)	60.70 USD	0.61 USD/g
Petroleum Ether (7.55 L)	19.60 USD	0.20 USD/g
Ethyl Acetate (3.02 L)	10.84 USD	0.11 USD/g
Methanol (6.04 L)	22.51 USD	0.23 USD/g
Stearic acid (C18:0) (38.31 g)	570.94 USD	5.71 USD/g
DME (4.41 L)	230.82 USD	2.31 USD/g
Total cost of main substrates and reagents	2013.70 USD	20.14 USD/g
(2) Various enzymes		
NmCSS (1.38 g)	484.40 USD	4.84 USD/g
PmST1(1.32 g)	91.30 USD	0.91 USD/g
PmPpA (1.34 g)	12.17 USD	0.12 USD/g
BLNahK (1.06 g)	38.92 USD	0.39 USD/g
AGX1 (0.79 g)	71.66 USD	0.72 USD/g
EcGalK (1.59 g)	69.27 USD	0.69 USD/g
AtUSP (1.38 g)	50.15 USD	0.50 USD/g
CjCgtA (2.65 g)	547.83 USD	5.48 USD/g
CjCgtB (3.18 g)	365.22 USD	3.65 USD/g
EGC-II E351S/D314Y (21.17 g)	2641.26 USD	26.41 USD/g
SA_SCD (8.82 g)	32869.75 USD	328.70 USD/g
Total cost of various enzymes	37241.93 USD	372.42 USD/g
(3) Labor cost	5519.68 USD	55.20 USD/g

(4) Purification cost	2434.79 USD	24.35 USD/g
In total	47210.10 USD	472.10 USD/g

Further engineering studies based on this strategy, such as immobilization of the enzymes and optimization of the purification steps, are feasible for greatly reducing the cost of the entire process. Therefore, we believe MOCECA provide a cost-effective means for the large scale preparation of ganglioside in the future.

Point #6: The biological tests mainly serve to show (general) biological activity of the products, which is fine. As a study on the neurobiological activity of each product a more extensive study would be warranted.

Response: Thanks for reviewer's helpful suggestions. In this study, our focus is to describe a highly modular chemo-enzymatic cascade assembly (MOCECA) strategy for customized and large-scale synthesis of ganglioside derivatives with various glycan and ceramide epitopes. Our cell experiments have demonstrated that these GM1 structures with different ceramides do indeed have diverse biological activities, laying the foundation for further research. It is very necessary to study the neurobiological activity of each product, but it is not the main topic of this manuscript. Benefiting from the diversified synthetic advantages of MOCECA strategy, molecular biology and cell biology methods will be used to conduct deeply and extensive research on the neurobiological activity and regulatory mechanisms of ganglioside analogues in future research.

Point #7: indicate under "methods" what is given in the SI. Or give a more detailed description of the contents of the SI under "additional information"

Response: Thanks for reviewer's helpful suggestions. According to your suggestion, we have added a more detailed description of the contents of the SI under "additional information" to explain the various experimental methods and data included in Supplementary information.

Point #8: line 394 replace "regiosomer" by "regioisomer"

Response: We thank the reviewer and have made this change as suggested.

Reviewer #4 (Remarks to the Author):

Point #1: Guangyu and coworkers report the synthesis of various ganglioside derivatives utilizing a chemoenzymatic cascade to assemble the individual fragments while employing a chemical synthetic approach towards making the ceramide component. The authors proceed to evaluate the neurobiological activities of these derivatives. This reviewer will not comment on this component of the manuscript and instead will focus on the synthesis. There are two notable synthetic advances here. One being the improved chemical synthesis of the ceramide component - previous work here has relied on particularly lengthy routes and ones that are far more expensive to carry through. The chemistry here is highly scalable and amenable to cost-effective synthesis of ceramide analogues. The second is the highly scalable nature of chemo-enzymatic cascade reactions. While the use of enzymes for assembling gangliosides have been known for a few decades now, these are rarely performed on such large scales. The authors make this point in their introduction without referencing any literature - please address this (line 83). These advances in themselves render this manuscript worthy of publication in Communications Chemistry.

Response: We are very grateful to the reviewer for the affirmation of the notable synthetic advances in this article. Through extensive browsing of research and review articles on ganglioside synthesis, we found that there have been few such large-scale studies as the manuscript, although the use of enzymes for assembling gangliosides have been known for a few decades now. We apologize for the oversight to cite representative literature. According to your suggestion, we have added corresponding references 1 and 23 to support the sentences in line 83.

Minor concerns:

Point #2: The supporting information is to a high-standard. Some small points here: Did the authors check for racemization at some point after forming the alpha-amino ketone intermediate? These often have issues with racemization and would require ee%

measurements to confirm enantiopurity. How did the authors confirm the relative stereochemistry resulting from the glycosylation steps?

Response: We gratefully appreciate for the Reviewer's valuable comment.

We have also realized the importance of the asymmetric reduction of enones **31** and **32**. We prepared the corresponding alcohols **33** and **34** according to Liebeskind's reported procedures (*Organic Letter*, 2007, 9, 2993), which we cite as ref. 17 in the SI. It's worth noting that the ketone reduction should be performed before the desilylation step with TBAF to avoid the racemization. This racemization problem is similar with Liebeskind's report (*Organic Letter*, 2007, 9, 2993). Additionally, we can further improve the diastereomeric purity by recrystallization of alcohols **33** and **34**. For the ketone reduction step, we have screened reducing reagents and reaction conditions, the bulky reducing agent lithium tri-tert-butoxyaluminum hydride ($\text{LiAlH}(\text{OtBu})_3$) was used as the optimal agent to exhibit high yield and diastereoselectivity in the reductive reaction step. This reagent has two advantages. Firstly, it only has one hydride to react with the ketone group of enones **31** and **32**, unlike LiAlH_4 , so it's a lot easier to control the reaction using the proper stoichiometry. Secondly, big bulky tert-butoxy groups help to modulate the reactivity and stereoselectivity. $\text{LiAlH}(\text{OtBu})_3$ selectively reacts with the ketone group of enones with other functional groups intact to give alcohols **33** and **34** in excellent yields and anti diastereoselectivities.

As for of the glycosylation steps, glycosynthases and glycosyltransferases play an important role in the stereoselectivity of anomeric configuration. we firstly take the preparation of glycosylsphingosines **6-10** for example. Actually, a mutant glycosynthase EGC-II E351X have been reported by Withers' group in 2006 for synthesis of gangliosides from glycosyl fluorides and sphingosine analogues (*J. Am. Chem. Soc.* 2006, 128, 6300). The work is now cited as ref. 12 in the SI. Based on Withers' results, we optimized the endoglycoceramidase II transglycosylation system in our work. Various fluoro-oligosaccharides including GM1-F, GM2-F, GM3-F, GD2-F, and GD3-F directly assembled with D-sphingosine (d18:1) to give glycosylsphingosines **6-10** with excellent beta stereoselectivity. The mutant enzyme catalyze glycosylation reactions with substrates to produce glycosylsphingosines with

beta stereoselectivity after the check of NMR data. Additionally, all glycosyltransferases PmST1, CjCgtB, CjCgtA, and CjCstII used in the work have been reported (please see the details in page S3), the catalytic characteristics of these glycosyltransferases has been known by far. Various monosaccharides are enzymatically assembled with excellent chemoselectivity and stereoselectivity.

REVIEWERS' COMMENTS:

Reviewer #1 (Remarks to the Author):

The Authors have satisfactorily answered all reviewers' comments. The resulting manuscript is stronger and suitable for publication, provided that two minor issues are addressed:

1) The use of the word analogue throughout the manuscript is not appropriate. It should be changed to "molecular species" or "species" or "lipofoms" (see Levery, Methods in Enzymology, DOI: 10.1016/S0076-6879(05)05012-3).

2) In Fig.4 legend, the post test used following one-way ANOVA to determine significance between individual groups should be stated.

Reviewer #3 (Remarks to the Author):

The authors have adequately revised the manuscript and addressed all comments of the reviewers. I have no further comments. I recommend to accept the revised manuscript for publication.

Reviewer #4 (Remarks to the Author):

The authors have addressed my concerns and the manuscript may be published without further corrections from this reviewer.

Point-by-point responses to the Reviewer's comments

Reviewer #1 (Remarks to the Author):

The Authors have satisfactorily answered all reviewers' comments. The resulting manuscript is stronger and suitable for publication, provided that two minor issues are addressed:

Responses to Reviewer#1:

Point #1: *The use of the word analogue throughout the manuscript is not appropriate. It should be changed to "molecular species" or "species" or "lipofoms" (see Levery, Methods in Enzymology, DOI: 10.1016/S0076-6879(05)05012-3).*

Response: Many thanks for the Reviewer's suggestion. According to the editor's opinion, we think the use of term analogue is more appropriate and accurate in this situation. Ganglioside analogues are used in the manuscript.

Point #2: *In Fig.4 legend, the post test used following one-way ANOVA to determine significance between individual groups should be stated.*

Response: We thank the Reviewer for the helpful guidance. According to your suggestion, we have added Newman-Keuls multiple comparison test following one-way ANOVA to determine significance between individual groups in Fig.4 legend.

The updated version was listed:

Significant differences among groups were determined using Newman-Keuls multiple comparison test following one-way ANOVA analysis in GraphPad Prism software.

Reviewer #3 (Remarks to the Author):

The authors have adequately revised the manuscript and addressed all comments of the reviewers. I have no further comments. I recommend to accept the revised manuscript for publication.

Responses to Reviewer#3:

Response: We greatly appreciate the reviewer's recommendation and all the helpful guidance.

Reviewer #4 (Remarks to the Author):

The authors have addressed my concerns and the manuscript may be published without further corrections from this reviewer.

Responses to Reviewer#4:

Response: We greatly appreciate the reviewer's recommendation and all the helpful guidance.